behaviour

killer whale, *Orcinus orca*, bioacoustics, call repertoire, Ross Sea, Antarctica

**Author for correspondence:**
Rebecca Wellard
e-mail: becwellard@gmail.com

# Cold call: the acoustic repertoire of Ross Sea killer whales (*Orcinus orca,* Type C) in McMurdo Sound, Antarctica

Rebecca Wellard[1,2], Robert L. Pitman[3,5], John Durban[4] and Christine Erbe[1]

[1]Centre for Marine Science and Technology, Curtin University, GPO Box U1987, Perth, Western Australia 6845, Australia
[2]Project ORCA, Perth, Western Australia 6026, Australia
[3]Antarctic Ecosystem Research Division, and [4]Marine Mammal and Turtle Division, Southwest Fisheries Science Center, National Marine Fisheries Service, 8901 La Jolla Shores Drive, La Jolla, CA 92037, USA
[5]Marine Mammal Institute, Oregon State University, 2030 SE Marine Science Drive, Newport, OR 97365, USA

RW, 0000-0001-8427-564X; RLP, 0000-0002-8713-1219; JD, 0000-0002-0791-8370; CE, 0000-0002-7884-9907

Killer whales (*Orcinus orca*) are top marine predators occurring globally. In Antarctic waters, five ecotypes have been described, with Type C being the smallest form of killer whale known. Acoustic recordings of nine encounters of Type C killer whales were collected in 2012 and 2013 in McMurdo Sound, Ross Sea. In a combined 3.5 h of recordings, 6386 killer whale vocalizations were detected and graded based on their signal-to-noise ratio. Spectrograms of the highest-quality calls were examined for characteristic patterns yielding a catalogue of 28 call types (comprising 1250 calls). Acoustic parameters of each call were measured and summarized by call type. Type C killer whales produced complex calls, consisting of multiple frequency-modulated, amplitude-modulated and pulsed components. Often, two components occurred simultaneously, forming a biphonation; although the biphonic components did not necessarily start and end together, with one component lasting over several others. The addition and deletion of components yielded call subtypes. Call complexity appears stable over time and may be related to feeding ecology. Characterization of the Type C acoustic repertoire is an important step for the development of passive acoustic monitoring of the diverse assemblage of killer whale ecotypes in Antarctica's rapidly changing marine ecosystems.

# 1. Introduction

Killer whales (*Orcinus orca*) are top predators distributed throughout the Earth's oceans. Although they are still considered to comprise a single species, studies have established that some populations demonstrate distinct morphological and genetic differences, social structures, diet preferences and acoustic repertoires [1–7].

In Antarctic waters, five killer whale ecotypes have been described: Type A, Type B (two forms: larger 'B1' and smaller 'B2'), Type C and sub-Antarctic Type D, with ecotypes displaying morphological [2,7,8] and genetic differences [9,10]. Emerging data further suggest ecological variability in habitat preferences, prey specialization and foraging behaviours [2,7,8].

Type C killer whales (also recognized as Ross Sea killer whales; [11]) are known mainly from summer sightings off eastern Antarctica [2], where they occur in continental shelf waters along the fast ice edge. They also occur deep into ice leads where they hunt for fish, such as the Antarctic toothfish (*Dissostichus mawsoni*) and additional fish species as documented by stable isotope analysis [12,13] and field observations [14–16]. Type C killer whales are, currently, the smallest killer whale form known (adult Type C males grow up to 6.1 m, in comparison to Type A males up to 9.2 m; [17]), and are easily identifiable by their dorsal cape and narrow, slanted eyepatch [2,18].

Type C killer whales have been commonly reported in McMurdo Sound, Ross Sea, eastern Antarctica for more than a century [2,16,19–23]. Annual sighting records since the 1970s indicate killer whale presence in the area soon after the seasonal icebreaking has begun. Killer whales are found to take advantage of foraging habitat made accessible when an icebreaker opens up the channel for supply ships to gain access to McMurdo Station, resulting in opportunities for data to be collected from these animals at close distances while they use the icebreaker channels [2]. Studies using tags and photo-identification indicate the Type C killer whales may in fact be summer residents in McMurdo Sound [16,24]. A photo-identification project identified 352 individual Type C killer whales in McMurdo Sound during seven summers from 2001–2002 to 2014–2015 and estimated an annual average population of 470 [16]. However, there are very few records of Type C killer whales in other seasons [25], resulting in knowledge gaps and uncertainty relating to year-round usage of the Ross Sea, in general, and the Ross Sea Region Marine Protected Area (RSRMPA).

The mobile nature of killer whales makes this species difficult to study by direct observation. While satellite telemetry [24,26] can track individuals for weeks to months, passive acoustic monitoring (PAM) is a technique used to detect occurrence and relative abundance in the longer term. Using autonomous recording systems in remote and isolated regions, such as the Ross Sea, would allow data to be collected year-round, independent of restricting polar conditions such as inclement weather, limited daylight and increased ice coverage. Quantitatively describing the acoustic repertoire of a species and potentially distinguishing sympatric ecotypes is an important first step for establishing effective PAM programmes. To describe and ascertain a repertoire and understand vocal behaviour, concurrent visual sightings and acoustic recordings need to occur, with sufficient visual sightings to identify not only species, but in this case, ecotype as well.

Studies of the vocal characteristics of different killer whale populations have identified a mix of unique and shared call types and documented 'vocal culture' whereby different killer whale groups exhibit distinct dialects [27–30]. These dialects are stable through time [31,32] and are a learned behaviour [27,33,34]. Groups with very similar repertoires have been shown to be more closely related than groups that share fewer calls, with some pods of related matrilines sharing many or all of the components in their repertoire [28,35,36]. Differences in calls among spatially separated populations of killer whales are apparent from studies worldwide [27,29,33,35,37] and have resulted in effective monitoring of these populations through the use of passive acoustic listening stations.

Like other delphinids, killer whales have an acoustic repertoire that consists of sounds fitting into three broad classes: clicks, whistles and burst-pulse sounds. Echolocation clicks are broadband pulses (10–100 kHz), less than 1 ms in duration, typically emitted in trains lasting several seconds; they are used for navigating and foraging [38–40]. Whistles are continuous tonal signals, which may be constant-wave (i.e. do not change frequency over time) or frequency-modulated. While the fundamental frequency is the lowest frequency, harmonic overtones may occur at integer multiples of the fundamental frequency [41]. If whistles are amplitude-modulated, sidebands occur [42,43]. Killer whale whistles range from less than 1 to up to several seconds in duration [35,44]. Whistles of killer whale populations in the North Pacific have fundamental frequencies ranging from 1 to 36 kHz [35,44–47], while eastern North Atlantic killer whale populations have fundamental frequencies up to

74 kHz [48]. Burst-pulse sounds are broadband sounds that consist of rapidly repeated pulses; and the pulse repetition rate (PRR) is higher than the click repetition rate in echolocation click trains. Due to this high PRR, individual pulses are mostly not resolved in Fourier analysis, and so, in spectrograms, burst-pulse sounds typically appear as frequency-modulated sounds with numerous sidebands that are related to the PRR. The PRR can always be read off the spectrogram as the 'harmonic interval' between neighbouring contours [42]. Frequency-modulation of the contours in burst-pulse sounds is related to changing PRRs. Killer whale burst-pulse sounds have most energy between 500 Hz and 25 kHz, and last 0.5–1.5 s [28,29,35,37,49–54]. Both whistles and burst-pulse sounds are believed to be communicative signals used in social contexts, functioning as contact signals in coordination of behaviour and in group identification [35,55,56].

Not all sounds fit into such distinct classes of whistles and burst-pulse sounds. Killer whales, as well as false killer whales (*Pseudorca crassidens*) and probably other cetaceans, further produce graded sound types that lie along a continuum from whistles to pulses [57,58]. This continuum is achieved by progressively increasing the amplitude modulation of whistles until successive pulses are formed, and increasing the inter-pulse interval thereafter. Whistles, burst-pulse sounds and graded types have been used reliably to categorize calls specific to killer whale populations [29,50,59].

Despite the reliability of sighting Type C killer whales in McMurdo Sound, acoustic studies on their call repertoire have been sparse and had their limitations. In McMurdo Sound, both Type B and Type C have been sighted and seen to use the ice channel [16,23], hence why confirming ecotype is pertinent for concurrent acoustic recordings. There have been four previous studies describing killer whale vocal behaviour in Antarctic waters [54,60–62], with only one report having concurrent visual data confirming ecotype Type C with certainty [61].

Previous reports briefly described the underwater sounds of killer whales in the Ross Sea [60,62,63], but no concurrent photographs could confirm the ecotype. Billon [62] described 18 distinct call types from recordings off Ross Island, McMurdo Sound, during one encounter presumed to be one group in the austral summer of 1979–1980. Richlen & Thomas [54] analysed recordings taken in 1979 along a lead within the fast ice in McMurdo Sound from a group of seven to nine killer whales. Seven discrete call types were identified, with Richlen & Thomas [54] reporting the acoustic repertoire similar to sounds described from fish-eating killer whale populations in other oceans, and suggesting a pod-specific repertoire due to the consistent repetition of call types. Due to insufficient diagnostic features on photographs from this encounter, confirmation of ecotype was not possible.

A recent study analysed opportunistic acoustic data collected concurrently with visual confirmation of a group of Type C killer whales near the Eckström Ice Shelf in the eastern Weddell Sea [61]. However, this study had limitations, such as a 15 kHz bandwidth, which meant the classification analyses in this study were restricted. Another limitation was the low encounter rate; only four killer whales were sighted during the acoustic recording. This low encounter rate of individuals may bias the described call repertoire. When describing the repertoire of a species or ecotype, the study would ideally maximize data representation and avoid oversampling specific groups or individuals. Acoustic data would ideally be collected from different groups and individuals, displaying a multitude of varying behaviours, to capture the potentially broad acoustic repertoire of the subject.

Our study conducted in McMurdo Sound is the first to extensively describe the call repertoire of confirmed Type C killer whales. Our detailed description of acoustic characteristics provides an initial step towards comparing and distinguishing Type C killer whale acoustics with those of other killer whale populations in the Southern Hemisphere and is essential for PAM to be effective when monitoring populations of sympatric ecotypes.

# 2. Methods

## 2.1. Study area and data collection

Acoustic data were collected near the fast ice edge in McMurdo Sound, Ross Sea, Antarctica, between December 2012 and January 2013 (figure 1). The water depth in McMurdo Sound ranges from a shallow 200–400 m slope on the west side to a 700 m deep trench on the east side [66]. Most of the data collection commenced with a scouting flight by a helicopter from McMurdo Station. Upon detecting killer whales, the helicopter landed on the fast ice at least 200 m from the ice edge. Killer whales usually occurred along the fast ice edge, but were sometimes found along leads in the fast ice or at isolated breathing holes, 0.5 km or more from the fast ice edge. The hydrophone was

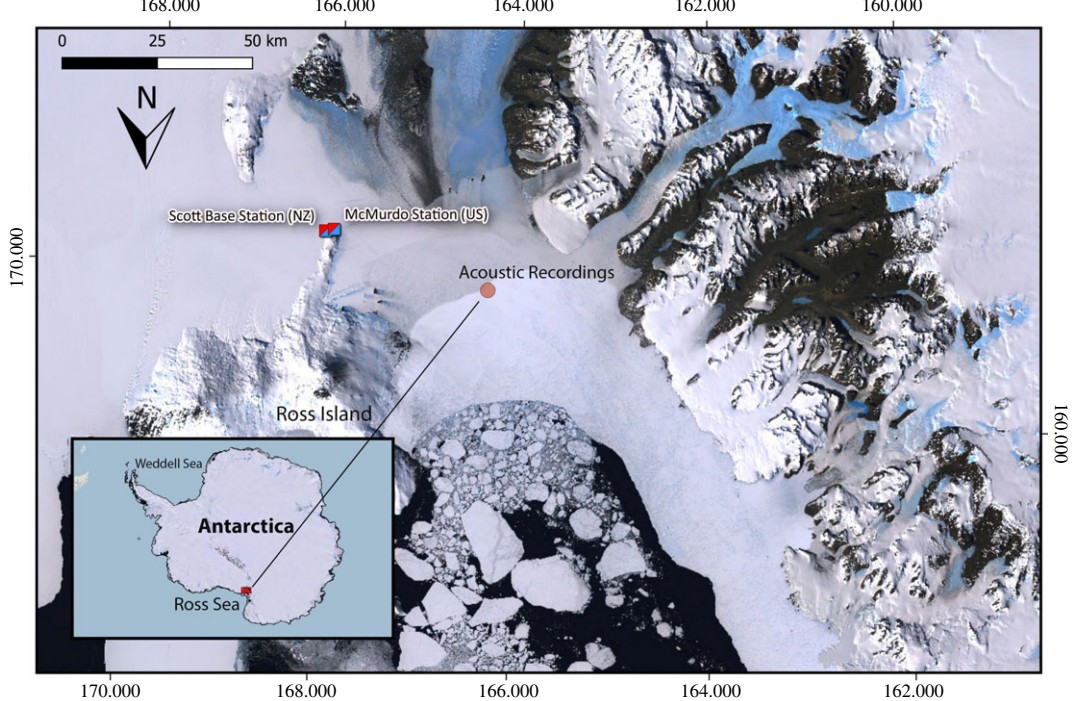

**Figure 1.** Map of Antarctica showing the location of the study area and marked point where acoustic recordings were taken in McMurdo Sound between December 2012 and January 2013. Map produced using QGIS mapping software [64] with Quantarctica package [65]. Satellite imagery provided by Norwegian Polar Institute based on Landsat satellite images from previous years does not reflect sea ice coverage during the 2012–2013 season.

hand-deployed into the water at the ice edge in the immediate vicinity of killer whales (i.e. less than 100 m range) approximately 3–4 m below the surface. Acoustic recordings were obtained using an M-Audio Microtrack 24–96 recording unit with a Lab-Core LAB-40 hydrophone system (approx. 5 Hz–85 kHz bandwidth) and a 20 dB in-line amplifier. Sound was sampled at 96 kHz, 24 bit, or 44.1 kHz, 24 bit, providing a bandwidth of 48 kHz and 22.05 kHz, respectively.

During the recording, information on killer whale ecotype, group composition, number of animals and behavioural state was noted. Individual whales were photographed as part of a photo-identification study [16]. Surface behaviour was assigned to one of four behavioural states, which were adapted from previous killer whale studies [35,38,67–69]: (i) travelling, (ii) foraging, (iii) milling/resting or (iv) socializing.

## 2.2. Acoustic analysis

Acoustic recordings were inspected both visually and aurally using acoustic software Raven Pro 1.5 [70]. A fast Fourier transform (FFT) was computed in Hann windows of 1024 and 512 samples with 90% overlap for the recordings sampled at 96 and 44.1 kHz, respectively, resulting in a frequency resolution of about 90 Hz and time window of about 11 ms. Only recordings made during a confirmed sighting of Type C killer whales were included in our analysis. Calls were visually graded based on their signal-to-noise ratio (SNR): Grade 1 (Poor) if the signal was faint, but still visible; Grade 2 (Average) if the signal was distinct and clear; and Grade 3 (Good) if the signal was strong and prominent. Only Grade 2 and 3 calls were selected for analysis.

## 2.3. Call categorization

Calls graded 2 and 3 were sorted into distinct categories to produce a call catalogue. These call categories were principally based on features that are readily discernible in spectrograms and demonstrate the unique aural characteristics of a call, such as the number of successive components (single or multi-component call), duration of the call, presence of simultaneous components (biphonic call) and the overall shape of the call's contour (figure 2). Calls were classified as biphonic if they had two

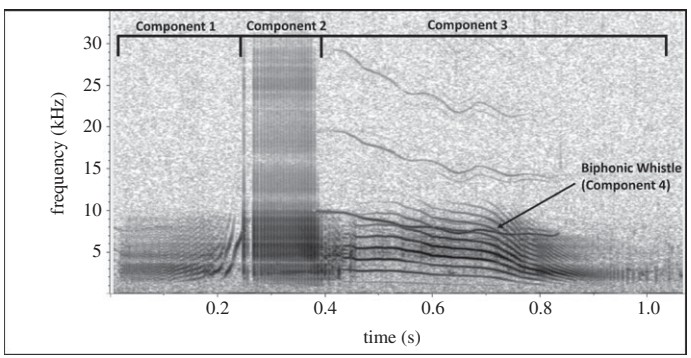

**Figure 2.** Spectrogram illustrating how calls are segmented into components. This is a four-component biphonic call categorized as call type McM1. Component 1 is a burst-pulse sound with the PRR increasing towards the end. The PRR can be read off the spectrogram as the sideband spacing (SBS). Component 2 is a series of single pulses. Components 3 and 4 make up a biphonation. Component 3 is a burst-pulse sound with an SBS of roughly 1 kHz. The PRR decreases towards the end. Component 4 is a whistle with harmonics that decrease in frequency over time ($f_s = 96$ kHz, 1024-point FFT, 90% overlap, Hann window).

simultaneous but independently modulated frequency components, otherwise they were classified as monophonic [49,71]. This methodology was based on previous studies using aural and spectrographic comparison for categorizing killer whale calls [27,31,72]. While some studies found that biphonations consist of a low-frequency component (LFC) and an upper-frequency component (UFC; e.g. [27]) or high-frequency component (HFC; e.g. [37,61]), we did not find this. In fact, in many biphonic calls recorded during this study, both components covered the same frequency band. Sometimes, one contour started out as the lower-frequency component, but increased in frequency over time, while the initially higher-frequency component ended lower than the former component. As our data did not confirm the LFC/HFC separation, we did not use this terminology.

Once all call categories had been examined, call categories that comprised fewer than three instances of that call were removed. Initial call categorization was conducted by two of the authors (R.W. and C.E.) and was subsequently confirmed by a test for interobserver reliability. Several call categories were found to have subtypes (i.e. variations of the primary call type). Subtypes were assigned if there were significant changes in a component's frequency contour, or an addition or deletion of one component [27,29] and at least three occurrences of the subtype.

## 2.4. Interobserver reliability test

To confirm the initial categorization of calls, a subset of 50 calls were randomly chosen and given to independent observers for classification [45,73–76]. Spectrograms were printed on individual sheets and shown in random order. Four observers with little to no acoustic analysis experience were asked to group the calls independently into an unspecified number of categories based on (i) call duration, (ii) the number of components, and (iii) similar contour modulations. A $\kappa$-statistic was then used to test for interobserver reliability [77]. $\kappa$ is a measure of difference, standardized to lie on a −1 to 1 scale. If the observers are in complete agreement in the classification of calls, then the Fleiss-Kappa statistic ($\kappa$) is equal to 1. If the agreement among observers is the same as would be expected by chance, then the $\kappa$ value is equal to 0. Negative values indicate agreement less than chance, i.e. potential systematic disagreement between the observers [78].

## 2.5. Call measurements

For each category, calls graded 2 and 3 were measured. For categories with 10 or fewer calls, all calls were measured. For categories with more than 10 calls, either 10 calls or 20% of calls were measured, whichever was greater. Of each call, up to 20 parameters were measured in Raven Pro 1.5 to quantify its spectro-temporal structure (electronic supplementary material, appendix S1 and table S1). Some of the parameters are more useful for quantifying broadband calls like burst-pulse sounds (e.g. entropy measures and quartile frequencies), while others are more useful for whistles (e.g. start, end, minimum and maximum frequencies of the fundamental contour).

Calls consisted of one or more components, with many calls consisting of both whistle and burst-pulse components. Some calls also had simultaneous biphonic components. For multi-component

royalsocietypublishing.org/journal/rsos　　R. Soc. open sci. **7**: 191228

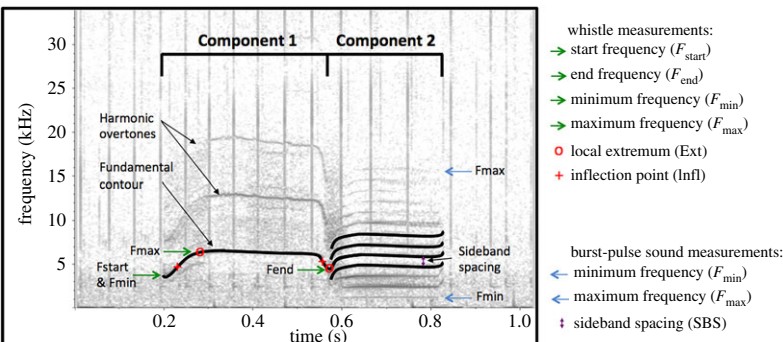

**Figure 3.** Spectrogram illustrating the parameters that were measured for acoustic analysis. This is a two-component call categorized as call type McM9. Component 1 is a whistle with harmonics and occasional weak sidebands indicative of amplitude modulation. Component 2 is a burst-pulse sound with a constant SBS of approximately 1 kHz. Parameters measured off the fundamental contour are illustrated: the start, end, minimum and maximum frequency measurements are labelled by arrows. The local extrema are denoted by a red circle. Inflection points along the contour are denoted by a red cross. The SBS is denoted by a double-ended arrow ($f_s = 96$ kHz, 1024-point FFT, 90% overlap, Hann window).

calls, parameters were measured separately for each component (figure 3). For whistles, measurements were taken off the fundamental contour; however, it was often easier to measure features off higher harmonics where the noise was less. Features such as duration, extrema, inflection points, frequency-modulation (FM) rate and steps are the same in harmonic overtones and fundamental. Frequency measurements such as start, end, minimum and maximum frequency are scaled; i.e. a factor $n$ higher for the $n$th harmonic (where the fundamental was counted as the 1st harmonic and the first overtone was considered the 2nd harmonic). For example, if measurements were done off the first overtone, then the measurements were divided by 2 in order to correspond to the fundamental.

As whistles have sinusoidal waveforms and burst-pulse sounds are series of rapid pulses, these two call types should be easily discernible from their waveforms. However, in recordings taken in the field, there is always interfering sound (due to ice noise, overlapping sounds from other killer whales in the group and recording artefacts) that makes it nearly impossible to distinguish sounds in recorded pressure time series. The different sounds that were recorded are more easily discerned in spectrograms, but recording (primarily, sampling frequency used, filters applied) and analysis settings (primarily, number of Fourier components) affect how well whistles and burst-pulse sounds can be told apart [42]. The pulses in burst-pulse sounds are typically too fast to be resolved in spectrographic analysis, and instead appear as frequency contours related to the PRR. All of the contours seen may occur at harmonic intervals, being integer multiples of a fundamental, making it impossible to tell whether the underlying call is a whistle with harmonics or a series of rapid pulses. In order to describe the different components of the recorded calls, we used the default from Watkins [42] calling sounds with fewer than five harmonics a whistle, and otherwise a burst-pulse sound. We also note that the majority of calls recorded transitioned gradually from burst-pulse sounds to whistles (or vice versa) by increasing the PRR and decreasing the inter-pulse interval until continuous tones were formed (or vice versa). Calls were divided into components not only when they transitioned from a whistle to a burst-pulse sound, or vice versa, but also when there were rapid shifts in the call's PRR [27].

## 3. Results

Acoustic recordings were collected during nine separate encounters with Type C killer whales that were confirmed to ecotype by diagnostic features (figure 4). Group size ranged from 8 to 125 individuals, including adults, subadults and calves (see electronic supplementary material, appendix S2 and table S1 for full encounter details). A summed total of 392 killer whales were counted over the nine encounters, although some of these individuals were re-sightings. A total of 167 distinct individuals were photographically identified from natural markings [16]. Most individuals were identified in only one sighting, yet 47 individuals were identified in multiple sightings. The maximum number of resights of the same animal was four (i.e. this animal was seen in four out of the nine encounters) on three different days.

Behaviour documented during acoustic recordings included all four behavioural states, with predominant behaviours of socializing, foraging and travelling observed. For most sightings, the killer

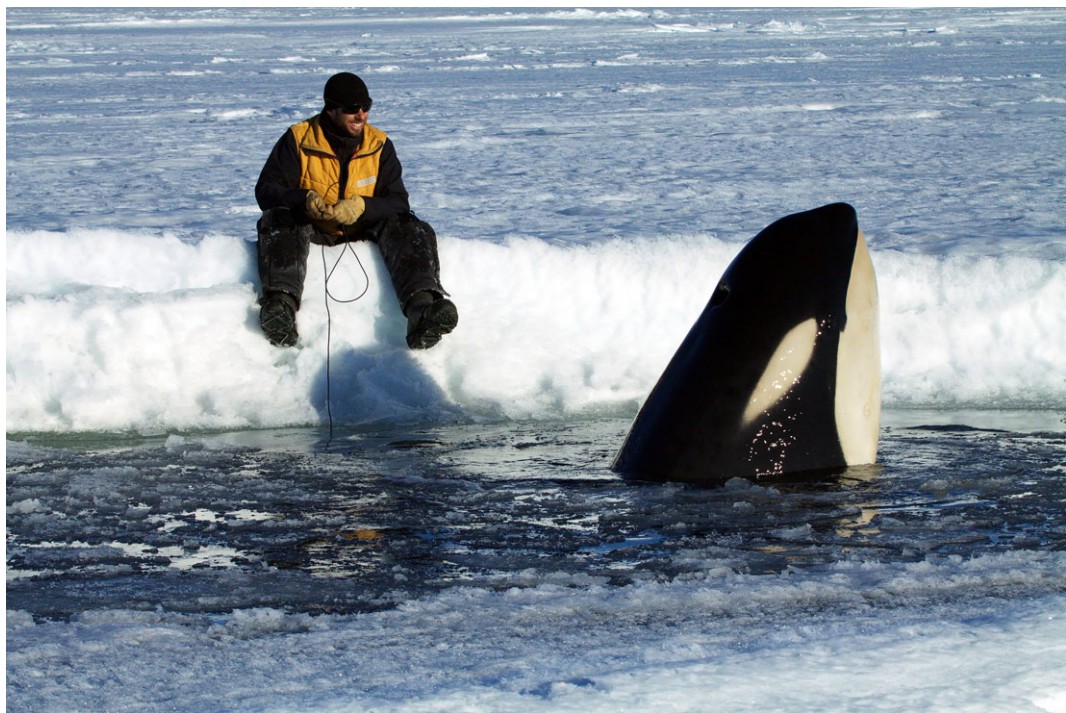

**Figure 4.** Photograph of a Type C killer whale encountered during acoustic recordings on 4 January 2013 in McMurdo Sound. The photograph shows the narrow, slanted eyepatch that is oriented at a 45° angle to the long axis of the body. Image by R.L.P.

whales were either travelling along the fast ice edge or foraging under the ice; i.e. they disappeared under the ice and often resurfaced in the same area several minutes later. Often, younger animals stayed at the surface near the ice edge while the adults foraged, so group behaviour often included simultaneous foraging and socializing. Type C killer whales, as a fish-eating ecotype, tend to aggregate into large groups, and therefore, it was difficult to discern stable constituent subgroups, probably matrilines.

A total of 3 h and 33 min of killer whale calls were analysed from 24 recordings and hydrophone deployments in McMurdo Sound resulting in 6386 killer whale vocalizations detected and subsequently graded. After removing Grade 1 calls, vocalizations were sorted into 35 call categories with six subtypes. Categories with fewer than three examples of each type were removed yielding a total of 1250 vocalizations placed into 28 categories, inclusive of four subtypes. Summary statistics for the acoustic parameters of each call type are listed in electronic supplementary material, appendix S1, table S2; a spectrogram of each call type with all parameters measured is given in the separate call catalogue (electronic supplementary material, Appendix S2).

## 3.1. Classification by human observers and interobserver reliability test

Call categorization by two experienced acousticians (R.W. and C.E.) yielded 28 categories. The visual inspection method conducted by four naive judges showed that observers agreed on the classification of the killer whale calls and the majority of calls were placed into the same categories by each observer, with a moderate level of agreement in classification of calls across the 28 categories (Fleiss-Kappa statistic, $\kappa = 0.515$, $z = 41.8$, $p < 0.0001$). These results show that clearly defined call types exist in the repertoire of Type C killer whales.

The most common call types were McM3, McM2, McM1, McM10, McM15, McM7 and McM5 ($n = 130$, 10.4%; $n = 111$, 8.9%; $n = 101$, 8.1%; $n = 95$, 7.6%; $n = 89$, 7.1%; $n = 88$, 7.0%; $n = 84$, 6.7%; respectively), while the other 21 call types comprised the remaining vocalizations analysed ($n = 552$, 44.2%). A total of four call categories were deemed subtypes. Three of the four subtypes (McM1a, McM5a and McM15a) had a deletion of one or more components from the primary call, while the remaining category (3a) had a variation in one of the components' frequency contour (figure 5).

The number of multi-component calls was higher ($n = 886$, 71%) than the number of single-component calls ($n = 364$, 29%), and 21 out of the 28 call categories consisted of multi-component calls, representative of the complexity of these signals (figure 6). Some subtypes appeared as a

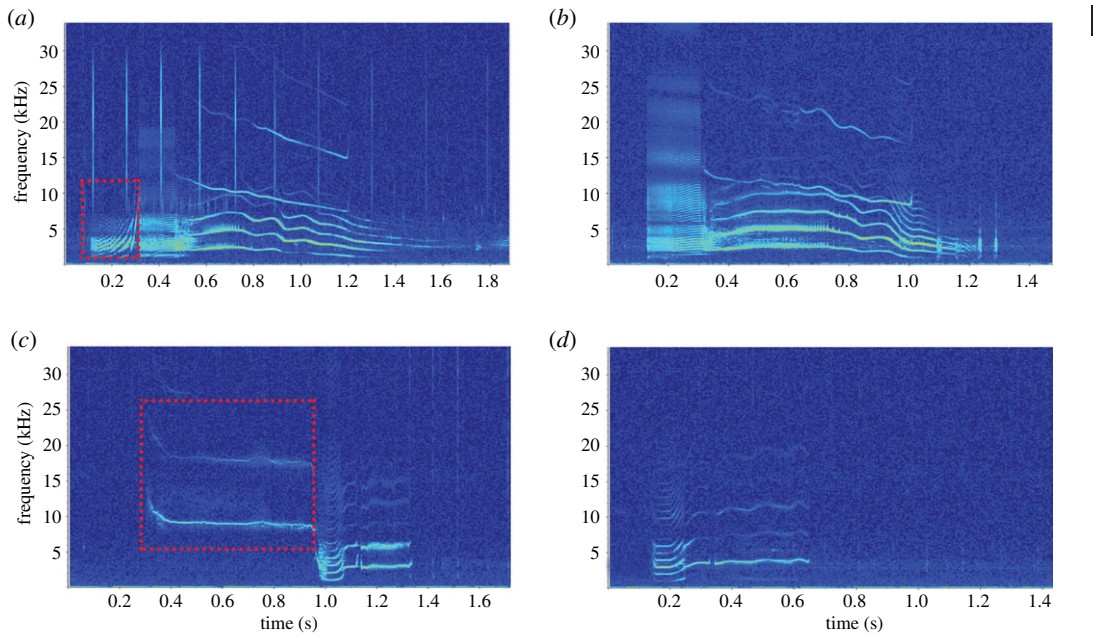

**Figure 5.** Spectrograms of call categories and the subtypes of these call categories recorded from Type C killer whales in McMurdo Sound: (*a*) call type McM1, a multi-component call with a biphonation, where the first component is highlighted in red. (*b*) Call type McM1a, a variation of call type McM1, where the first component is missing. (*c*) Call type McM15, a multi-component call, with the first component highlighted in red. (*d*) Call type McM15a, a variation of call type McM15 where the first component is missing ($f_s$ = 96 kHz, 1024-point FFT, 90% overlap, Hann window).

patterned sequence of 10 or more components (e.g. McM10; electronic supplementary material, appendix S2).

Of the 28 call categories, 46% were biphonic call categories ($n = 13$) and 54% were monophonic call categories ($n = 15$). In total, there were 532 biphonic calls measured and analysed. All biphonic calls had two components or more (figure 7). All biphonations consisted of a whistle and a burst-pulse sound. We note that if a biphonation occurred, the simultaneous components did not necessarily start and end at the same time. Quite often, the whistle in the biphonation (i.e. the biphonic whistle) covered multiple other components (e.g. Figure 7*b*,*e*; McM1, McM2, McM3a, McM4, McM18, McM19, McM22 in electronic supplementary material, appendix S2).

## 3.2. Call comparison with killer whale call repertoires described elsewhere in the Southern Hemisphere

All call types in this study were compared with other call types described in acoustic studies on repertoires of killer whales in the Southern Hemisphere. Out of the 28 call categories described here, seven call types were similar to calls previously documented. Call type McM1a, a multi-component and biphonic call starting with a series of pulses and followed by a burst-pulse sound with a biphonic whistle, has the same components and biphonation as call 'AM1' in Richlen & Thomas [54], and call '1' in Schall & Van Opzeeland [61] (figure 8*a*). Both singular and multi-component call types McM4, McM5, McM5a and McM14 were similar in structure to call types $B_7$, $A_2$, $A_{18}$ and $F_1$, respectively, described by Billon [62] (figure 8*b*–*d*,*g*). Call type McM7, a whistle with high frequency-modulation, is notably similar to call 'BC01' in Wellard *et al*. [58] and call 'AM4' in Richlen & Thomas [54] (figure 8*e*). Call type McM8, a two-component biphonic call with a burst-pulse sound and highly frequency-modulated biphonic whistle, is strikingly similar to call 'AM7' described by Richlen & Thomas [54] (figure 8*f*).

## 4. Discussion

This study yielded the first comprehensive description of the call repertoire of Type C killer whales, based on a unique dataset comprising concurrent visual and acoustic recordings that confirmed the ecotype over nine encounters with a summed total of 392 individuals. Previous reports and studies on

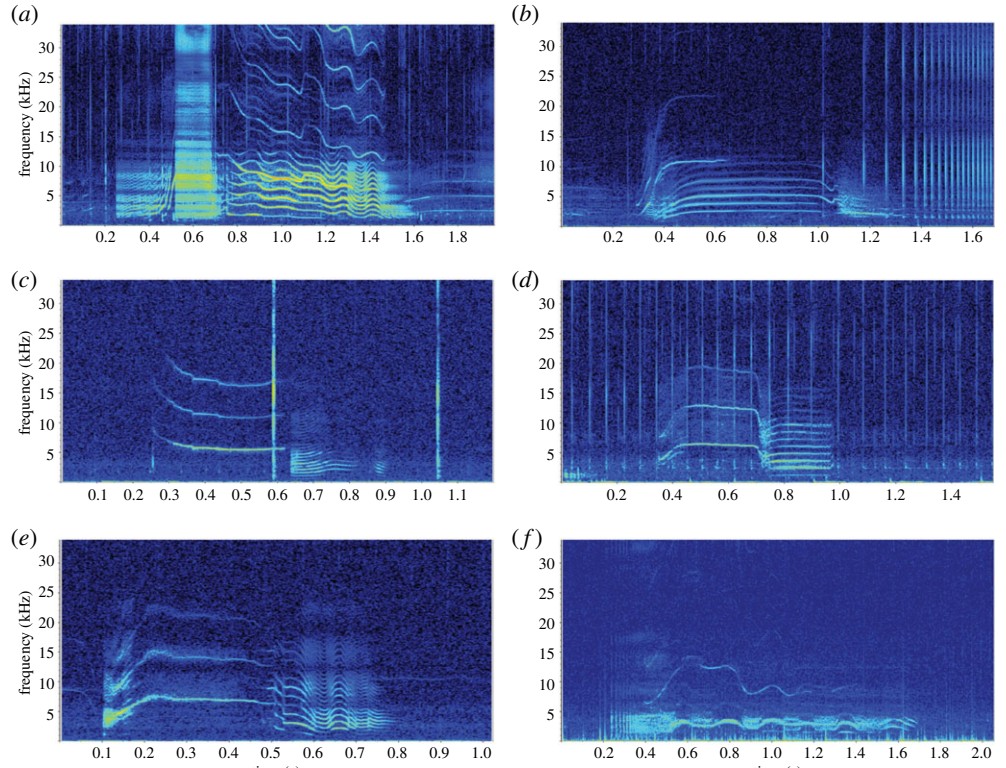

**Figure 6.** Spectrograms of multi-component calls recorded from Type C killer whales in McMurdo Sound: (*a*) call type McM1, with a biphonation; (*b*) call type McM3, with a biphonation; (*c*) call type McM5; (*d*) call type McM9; (*e*) call typeMcM10; and (*f*) call type McM18 ($f_s$ = 96 kHz, 1024-point FFT, 90% overlap, Hann window).

purported Type C killer whale vocalizations were limited in their visual confirmation of ecotype [54,62], in the scope of their acoustic analysis [21,60,63], and in the number of groups and individuals recorded [61]. This study with its larger sample size of groups and individuals and concurrent visual and acoustic observations delivered baseline data for identifying this killer whale ecotype using PAM systems and provides a foundation for future acoustic comparisons between sympatric Antarctic killer whale ecotypes.

## 4.1. Categorizing calls and vocal repertoire

A total of 28 call categories were described, inclusive of four subtypes being variations of the primary call type. This large number of call types is comparable to the repertoire of Type C killer whales off the Eckström Ice Shelf, eastern Antarctica, reported by Schall & Van Opzeeland [61], which comprised 26 call types from one encounter. The Type C repertoire of 28 call types is large in comparison to the seven call types of killer whales in one Ross Sea encounter in 1979 by Richlen & Thomas [54]. It is also larger than repertoires described for Northern Hemisphere killer whale ecotypes which range from 4 to 17 call types [29,32,37,50,59,79]. However, separating into call types is subjective.

A large vocal repertoire may reflect the feeding ecology of this ecotype or the behavioural state during the recording, or possibly both factors. Previous research has shown there is a clear distinction between call repertoires of mammal-eating killer whales and fish-eating killer whales, with the former producing fewer complex calls and exhibiting long periods of silence, and most vocal activity occurring only after marine-mammal kills and during social interactions [59,73,80,81]. Fish-eating killer whales are known to produce sounds prolifically in all behavioural contexts [38,82,83], probably because their prey has reduced hearing sensitivity at the frequencies of killer whale calls. By contrast, mammal-eating killer whales prey upon other whales, dolphins and pinnipeds with advanced underwater hearing abilities within the frequency range of killer whale calls, demonstrating that prey probably shape the vocal behaviour of the predator [59]. Type C killer whales are known to feed primarily on fish, and, similar to the fish-eating killer whales in the Northeast Pacific [50], their call repertoire displays a high number of call types and acoustic variability between call types. Behavioural context may also

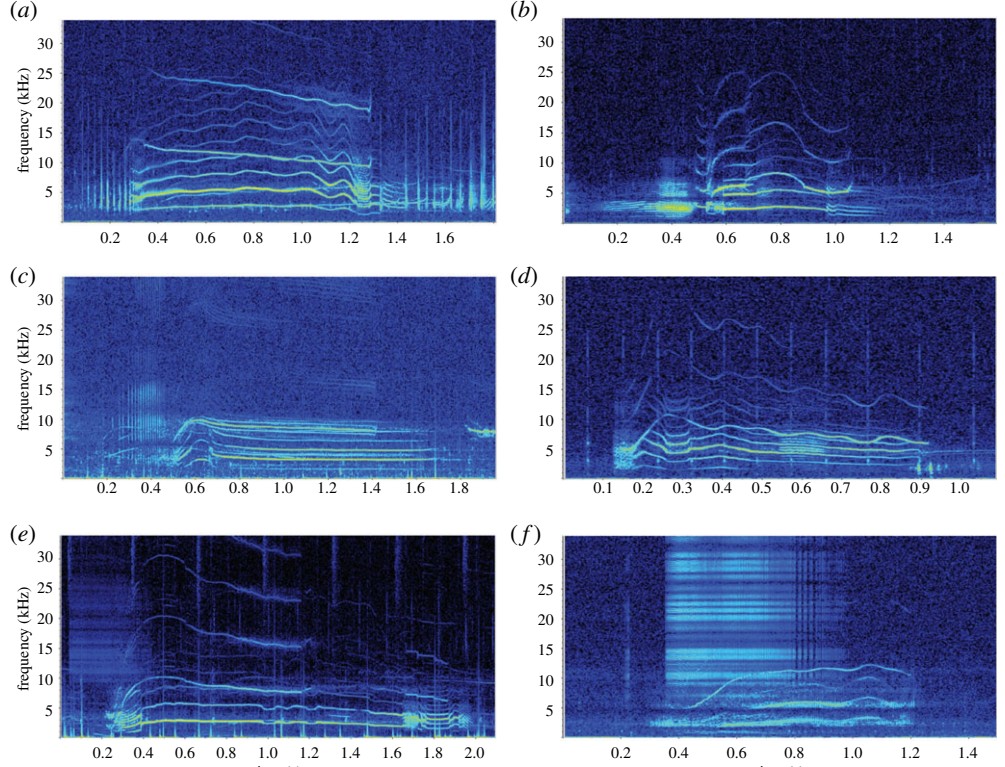

**Figure 7.** Spectrograms of biphonic calls recorded from Type C killer whales in McMurdo Sound: (*a*) call type McM2, a multi-component biphonic call, with distinct harmonics in the biphonation. (*b*) Call type McM3a, a multi-component biphonic call; this call is also a variation of call type McM3 where the first pulse starts well before the biphonic whistle commences. (*c*) Call type McM4, a multi-component biphonic call, with harmonics and weak sidebands in the biphonation. (*d*) Call type McM8, a biphonic call, with distinct harmonics in the biphonation. (*e*) Call type McM22, a multi-component biphonic call, with the biphonation evident at the start of the call at a low start frequency. (*f*) Call type McM23, a multi-component biphonic call ($f_s$ = 96 kHz, 1024-point FFT, 90% overlap, Hann window).

influence call rate and call variability within killer whale repertoire. Previous studies have reported an increase in the production of call types and call rate during observed social and foraging behaviour and a lower call rate during travelling [35,75,84]. The most common killer whale behaviours observed in this study were travelling and foraging under the ice and socialising at the surface, which could account for an increase in call rate and call variation. Both factors of feeding ecology and behavioural context need to be considered when examining call repertoire and using PAM technologies for detection.

## 4.2. Complexity of calls

The majority of call types described in this study are multi-component (68%), with many calls containing transitions from burst-pulse sounds to whistles (or vice versa). This is analogous to call types described by Richlen & Thomas [54] and Schall & Van Opzeeland [61], with many call types containing multiple components and transitions across components. Most interestingly, 39% of the call types described here start with a series of broadband pulses, which is akin to the one-third of call types observed by Richlen & Thomas [54] and the two-thirds of call types described by Schall & Van Opzeeland [61]. Such an acoustic feature should be considered when describing and identifying killer whale ecotypes in Antarctic waters, as this may serve as an acoustic marker for ecotype-identification when using remote listening stations.

A large percentage (46%) of call categories in this study contained biphonations. A biphonation appears as two independent but simultaneous contours in a call spectrogram [71,85] and has been described across a variety of mammal taxa including primates [86,87], canids [71,88,89] and cetaceans [90]. While the function of biphonation in calls is not understood, its occurrence in the vocalizations of different species implies a potentially important communicative role. Biphonic calls have been observed in the repertoire of fish-eating killer whales in both the Northwest Pacific [49] and Northeast Pacific [32], with these calls more common when animals occurred in mixed groupings consisting of

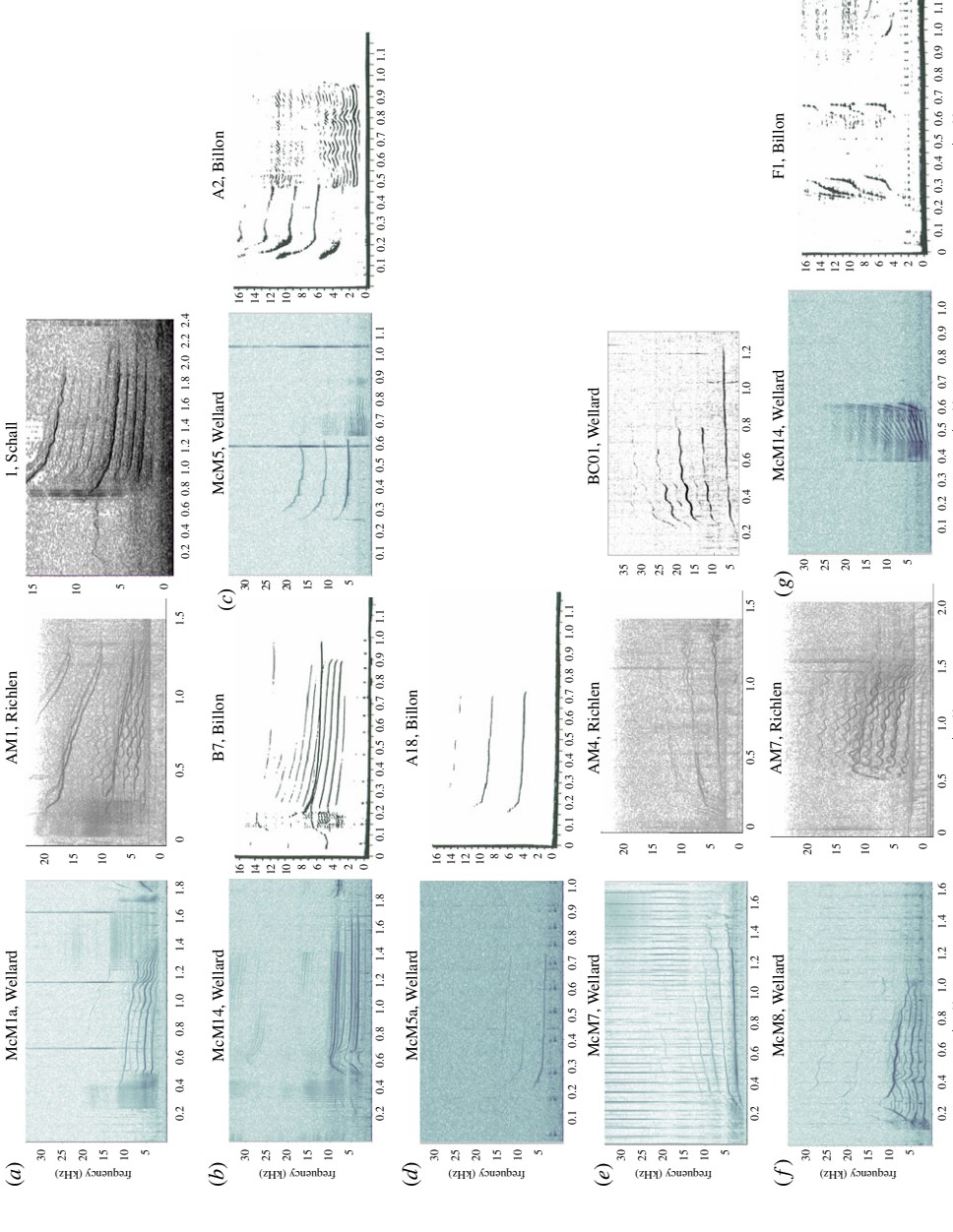

**Figure 8.** Spectrograms of call types recorded from Type C killer whales in McMurdo Sound (left panel $f_s = 96$ kHz, 1024-point FFT, 90% overlap, Hann window) compared to similar call types recorded from killer whales in the Southern Hemisphere: (*a*) Call type MdM1a compared with (from left) AM1 call type [54] and 1 call type [61]. (*b*) Call type MdM5 compared with call type A₂ [62]. (*d*) Call type MdM5a compared with call type A₁₈ [62]. (*e*) Call type MdM7 compared with (from left) AM4 call type [54] and BC01 call type [58]. (*f*) Call type MdM8 compared with (from left) AM7 [54] and (*g*) Call type MdM14 compared with F₁ [62].

members of different pods. This suggests that group composition influences the usage of such calls and that biphonic calls are possibly employed as markers of pod and matriline affiliation. Given the physical habitat at McMurdo Sound, characterized by a limited number of breathing holes, numerous family groups may be present within close vicinity, although information on the social structure of Type C killer whales is limited and it is unknown whether this ecotype is organized in stable groups similar to the matrilineal groups of Northern Hemisphere fish-eating killer whales. Pitman *et al.* [16] specifically mentioned a feeding area, described as the 'L' on the western side of the Sound, where Type C killer whales regularly gathered to feed under the ice. This fast ice habitat appears to affect killer whale groups aggregating around restricted feeding and breathing locations, which is demonstrated in this study where large groups of killer whales were encountered during recordings, with one encounter having up to an estimated 125 individuals (see electronic supplementary material, appendix S2 and table S1 for full encounter details). These large group sizes suggest that numerous family groups were present during recordings, which may explain a higher rate of biphonic calls used to locate group members.

The frequent use of biphonic calls could also be related to the shifting and changing habitat in McMurdo Sound. For killer whales, it was suggested that differences in the directionality of the components in biphonic calls can provide information on the orientation of a caller relative to a listener [91]. In McMurdo Sound, 4–5 m wide ice leads can rapidly close with changing wind and weather conditions, closing killer whale habitat for miles; and breathing holes, if they exist, can be kilometres apart. Calves were also observed during our study left at the ice edge when the adults were foraging under the ice. Hence, constant communication may be warranted to convey information on breathing holes and open leads between conspecifics. It is possible that animals use this directionality feature of biphonic calls to identify the signaller's orientation relative to the listener's position and communicate among individuals the shifting location of the ice edge and breathing holes.

## 4.3. Qualitative versus quantitative: categorizing killer whale calls

Due to the complexity of calls having multiple, successive and simultaneous components, simple quantitative techniques to group calls based on a set of frequency and time measurements were inapplicable, and therefore, calls were categorized manually by their spectrographic features and aural characteristics. Such manual classification is a common technique that has been used in numerous other studies on delphinid (including killer whale) call repertoires [27,28,35,74,81,92–94]. Some studies managed to validate perceptual classifications by less subjective techniques such as *k*-means testing and cluster analysis [58,95,96]. In order to increase the confidence in the categories of Type C killer whale calls, we employed two, instead of one, experienced bioacousticians for manual classification and assigned certainty via an interobserver reliability test with four additional scientists. Nonetheless, this method still has its limitations and is inherently subjective, with reduced reproducibility and criteria for categorization not clearly being defined. Currently, there is no defined technique for classifying killer whale call types, let alone for determining how many categories there should be, nor is there a singular technique for validating classifications. We measured a large number of features of each component in each call (electronic supplementary material, appendix S1 and table S1), for these data to be available for future, more quantitative and comparative studies on other Antarctic killer whale ecotypes.

## 4.4. Comparison with killer whale calls described elsewhere in the Southern Hemisphere

A comparison of call types showed that seven call types from this study had similar aural and structural characteristics to call types found in other studies on Southern Hemisphere killer whale repertoires. Similar calls have been noted in the call repertoire recorded off the Eckström Shelf [61], off the south coast of Western Australia [58] and in McMurdo Sound [54,62]. Of these seven call types, one call type from Wellard *et al.* [58] and one call type from Schall & Van Opzeeland [61] matched, which could be due to chance or similarity of the species' repertoire, rather than type. The limited similarity of calls between those reported here and confirmed ecotype Type C recordings by Schall & Van Opzeeland [61] off the Eckström Shelf may be due to limited sampling of individuals and behaviours in the Schall study, or may reflect geographical variation in vocal repertoire, with the Eckström Shelf being located on the opposite side of the Antarctic Continent from McMurdo Sound.

The remaining calls that matched were from McMurdo Sound, with four calls matched from Billon [62] and three calls from Richlen & Thomas [54]. Interestingly, the three call types that matched calls

described by Richlen & Thomas [54] in McMurdo Sound resulted in 33% of all Richlen calls matching with this study's catalogue. Antarctic killer whale ecotypes had not been described when Richlen & Thomas [54] and Billon [62] collected their recordings, and the few photographs that they took at the time do not show diagnostic features; however, our findings support the hypothesis that Richlen & Thomas [54] and Billon [62] did in fact record Type C killer whales between 1979 and 1982.

## 4.5. Call stability in the Ross Sea killer whale repertoire

There are a limited number of studies investigating the stability and temporal changes in killer whale call repertoires due to the lack of available long-term data from the same matriline or isolated population. A total of seven call types as described by Billon [62] and Richlen & Thomas [54] from killer whale calls recorded in McMurdo Sound between 1979 and 1982 were found to be similar in structural characteristics with call types described in this study suggesting call stability is present in the killer whale population found in McMurdo Sound, should these recordings also be of Type C ecotype.

Call stability has been reported in killer whale populations in the North Pacific, where call structure has remained stable for decades. Ford [28] first documented stability of a killer whale call repertoire by comparing recordings from the 1980s to historical recordings in the 1950s. This call stability was further validated when Foote et al. [32] found the repertoire and relative call usage was stable in southern resident killer whales described previously by Ford [28] for a further two decades. Deecke et al. [33], Riesch et al. [45] and Wieland et al. [97] also reported on call stability in northern and southern resident killer whales, describing minor structural changes over time, but the overall contour of the calls remained stable over tens of years.

The vocal repertoire of killer whales is thought to be a learned behaviour, rather than genetically coded [28,33,98], which can lead to the formation of dialects in sympatric populations and geographical variation in distant populations. Based on these findings, we hypothesize that Type C killer whales in McMurdo Sound, Ross Sea, may have a distinct dialect and a stable call repertoire. Further comparative acoustic research is needed to support the hypothesis.

## 4.6. Application to conservation management

The RSRMPA is the largest high-seas protected area in the world. The first of its kind in international waters, it was established in 2016 by the Commission for the Conservation of Antarctic Marine Living Resources (CCAMLR) following scientific advice [99]. Human presence in the Antarctic is increasing with an almost exponential increase in the annual number of visitors over the last two decades raising concerns about potential impacts on the Antarctic ecosystem [100]. Monitoring is needed to inform management of resources, determine the health of the ecosystem and assess whether the RSRMPA is achieving the objectives for which it was established.

The remoteness of the Ross Sea makes access for ecosystem monitoring difficult. The utility of upper-trophic-level species, or 'top predators', as ecosystem indicators and their effectiveness in informing management have been studied and discussed intensively [101–105]. These top-level predators can serve as indicators of change within the broader ecosystem of which they are a part. Understanding the movements, relative abundance and habitat usage of top predators, such as the killer whale, can help to assess the RSRMPA and assist in the development of policies and management decisions.

Autonomous acoustic recorders are an economical tool for long-term monitoring of vocalizing marine species such as killer whales, particularly in restricted locations and during prohibitive weather when vessels cannot go to sea—a problem during the Antarctic winter season. But PAM is not without its limitations. One major limitation is that animals must be vocalizing to be detected. Hence, understanding not only their call repertoires, but also their calling behaviour (i.e. call rates and behavioural context) is important for monitoring species using passive acoustics.

This study catalogued 28 complex and clearly recognizable call types of Type C killer whales, and, while the catalogue may not be complete, the number of call types and subtypes, along with the large number of individuals encountered, suggests a good portion of the repertoire may have been captured. Future research should investigate the call repertoire of other Antarctic killer whale ecotypes and examine the potential acoustic divergence. Identifying ecotype-specific dialects in the Antarctic region, in combination with genetic data, may also help determine matrilines and family groups, along with gaining a better understanding of cultural evolution and phylogenetic relationships within the most diverse killer whale community known.

Ethics. All field research was conducted under Antarctic Conservation Area Permit 2009-013, and Marine Mammal Protection Act Permit 14097 issued to NOAA Fisheries, Southwest Fisheries Science Centre.

Data accessibility. All data are presented in electronic supplementary material. Raw data are available in the Dryad Digital Repository: http://dx.doi.org/10.5061/dryad.37pvmcvfr [106].

Authors' contributions. R.W. analysed the recordings, categorized vocalizations, performed statistical analysis, prepared figures and tables, constructed call catalogue, drafted the manuscript. R.L.P. and J.D. planned and carried out data collection, provided revisions to the scientific content of the manuscript. C.E. assisted with analysis, categorized vocalizations, guided the writing of the manuscript, provided revisions to the scientific content of the manuscript.

Competing interests. We declare we have no competing interests.

Funding. Part of this work was supported by the US National Science Foundation under a grant to Stacy Kim (grant no. ANT-0944747); R.W. was the recipient of an Australian Postgraduate Award and an Australian Acoustical Society Education grant supporting data analysis and publication.

Acknowledgements. We would like to acknowledge the staff at McMurdo Station for their enthusiastic support, especially the helicopter pilots and the ice safety crew. For field assistance, we thank D. Mahon. We would like to thank Holly Fearnbach for conducting the photo-identification analysis for this study. We would like to acknowledge Aquatic Mammals for permission to incorporate published works into our figure 8 in this manuscript. We thank the four observers who participated in the interobserver reliability test.

Disclaimer. Any opinions, findings and conclusions or recommendations expressed in this material are those of the authors and do not necessarily reflect the views of the National Science Foundation.

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
