## [Reviewer comments · Royal Society Open Science]

Review History

RSOS-191228.R0 (Original submission)

Review form: Reviewer 1

Is the manuscript scientifically sound in its present form?

Yes

Are the interpretations and conclusions justified by the results?

No

Is the language acceptable?

Yes

Do you have any ethical concerns with this paper?

No

Have you any concerns about statistical analyses in this paper?

No

Recommendation?

Major revision is needed (please make suggestions in comments)

Comments to the Author(s)

Please see attached file with comments (Appendix A).

Review form: Reviewer 2**Is the manuscript scientifically sound in its present form?**

Yes

Are the interpretations and conclusions justified by the results?

Yes

Is the language acceptable?

Yes

Do you have any ethical concerns with this paper?

No

Have you any concerns about statistical analyses in this paper?

No

Recommendation?

Accept with minor revision (please list in comments)

Comments to the Author(s)

This manuscript describes the vocal repertoire of Antarctic Type C killer whales from data involving concurrent visual sightings. Access to confirmed and published call examples associated with marine mammal species are sometimes difficult to come by since the studies often lack findings beyond simple examples of the calls recorded. However, such findings can provide a reference for future studies and thereby avoid unnecessary duplication.

Beyond more minor edits to improve clarity and consistency, I feel that some restructuring is needed in order to avoid misleading readers and to better detail the methods and results. Mainly, the inclusion of the table of measured parameters suggests that the methodology used was more objective than it actually was. I think these findings are therefore more suited to the supplemental material than the main text. This manuscript could also be supported by an additional table to provide more details about the circumstances under which recordings were made.

More specific comments can be found in the attached word document (Appendix B), broken up by section.

Decision letter (RSOS-191228.R0)

08-Sep-2019

Dear Ms Wellard,

The editors assigned to your paper ("Cold Call: The Acoustic Repertoire of Ross Sea Killer Whales (Orcinus orca, Type C) in McMurdo Sound, Antarctica") have now received comments from reviewers. We would like you to revise your paper in accordance with the referee and Associate

Editor suggestions which can be found below (not including confidential reports to the Editor). Please note this decision does not guarantee eventual acceptance.

Please submit a copy of your revised paper before 01-Oct-2019. Please note that the revision deadline will expire at 00.00am on this date. If we do not hear from you within this time then it will be assumed that the paper has been withdrawn. In exceptional circumstances, extensions may be possible if agreed with the Editorial Office in advance. We do not allow multiple rounds of revision so we urge you to make every effort to fully address all of the comments at this stage. If deemed necessary by the Editors, your manuscript will be sent back to one or more of the original reviewers for assessment. If the original reviewers are not available, we may invite new reviewers.

- Data accessibility

<http://datadryad.org/submit?journalID=RSOS&manu=RSOS-191228>

- Competing interests

- Authors' contributions

All submissions, other than those with a single author, must include an Authors' Contributions section which individually lists the specific contribution of each author. The list of Authors should meet all of the following criteria; 1) substantial contributions to conception and design, or

acquisition of data, or analysis and interpretation of data; 2) drafting the article or revising it critically for important intellectual content; and 3) final approval of the version to be published.

- Acknowledgements

- Funding statement

on behalf of Dr Ari Friedlaender (Associate Editor) and Kevin Padian (Subject Editor)
openscience@royalsociety.org

Associate Editor's comments (Dr Ari Friedlaender):

To the Authors,

This study presents information on the acoustic calls and sounds of killer whales in the Ross Sea. There is a good amount of new information relating to the acoustic recordings as well as ancillary data that is useful for helping to contextualize some of the acoustic behavior of the species. Based on the comments from expert reviewers however, there are significant modifications that are required before the study can be considered for publication. As the paper focuses on acoustic information, the reviewers suggest substantial work is required in the methods section to better and more accurately describes various aspects of both data collection and analyses. These are critical in order for the findings to be validated. For example, the use of a hydrophone that was limited in its range to well below the known vocal range of the species could significantly affect the way in which lower frequency calls are presented and could bias the frequency range estimated for the species. This and other examples are shared by the reviewers and require attention. I hope you will take the opportunity to consider that being able to address these concerns should make the manuscript stronger and provide better justification for the conclusions whether they remain as is or require some modification based on limitations of the methods.

Please let me know if you have any questions.

Thank you.
Ari S. Friedlaender

Comments to Author:

Reviewers' Comments to Author:

Reviewer: 1

Comments to the Author(s)

Please see attached file with comments.

Reviewer: 2

Comments to the Author(s)

This manuscript describes the vocal repertoire of Antarctic Type C killer whales from data involving concurrent visual sightings. Access to confirmed and published call examples associated with marine mammal species are sometimes difficult to come by since the studies often lack findings beyond simple examples of the calls recorded. However, such findings can provide a reference for future studies and thereby avoid unnecessary duplication.

Beyond more minor edits to improve clarity and consistency, I feel that some restructuring is needed in order to avoid misleading readers and to better detail the methods and results. Mainly, the inclusion of the table of measured parameters suggests that the methodology used was more objective than it actually was. I think these findings are therefore more suited to the supplemental material than the main text. This manuscript could also be supported by an additional table to provide more details about the circumstances under which recordings were made.

More specific comments can be found in the attached word document, broken up by section.

Author's Response to Decision Letter for (RSOS-191228.R0)

See Appendix C.

RSOS-191228.R1 (Revision)

Review form: Reviewer 1

Is the manuscript scientifically sound in its present form?

Yes

Are the interpretations and conclusions justified by the results?

Yes

Is the language acceptable?

Yes

Do you have any ethical concerns with this paper?

No

Have you any concerns about statistical analyses in this paper?

No

Recommendation?

Accept with minor revision (please list in comments)

Comments to the Author(s)

In this revision, Wellard et al. have made substantial effort to address reviewer comments and edits. I have only a few very minor additional comments, and think that the study is suitable for publication.

Introduction

Lines 106-110 – This could possibly be restructured a bit, the way it currently reads sounds like graded call are a phenomenon unique to killer whales and false killer whales, but it's likely that it occurs in many of the odontocetes, we just haven't done the research yet to test that. For example, there is some evidence of graded call structures in wild populations of pilot whales.

Methods

Lines 157-158 – note the number of encounters/recordings for each of the sampling rates. Based on your explanation, it sounds like n=1 encounter at 44 kHz sampling rate and n=8 encounters at 96 kHz, is that correct? Add a sentence here explaining how you controlled for differences in the sampling rate, or explaining why that was not necessary.

Discussion

367 – 167 individuals were reported in the results, should be changed here to match.

Review form: Reviewer 2

Is the manuscript scientifically sound in its present form?

Yes

Are the interpretations and conclusions justified by the results?

Yes

Is the language acceptable?

Yes

Do you have any ethical concerns with this paper?

No

Have you any concerns about statistical analyses in this paper?

No

Recommendation?

Accept with minor revision (please list in comments)

Comments to the Author(s)

This version of a manuscript describing the vocal repertoire of Antarctic Type C killer whales has been much improved. However, I still see a few slight necessary changes, see details below. Particularly the legibility of the figure values is of concern, even if the figures are provided with 300 dpi.

Line 52: First figure mention – numbering should start with “1” here, suggest changing the order.

Line 91-92: The reference “42” talks about pulsed calls, not amplitude-modulated whistles. This needs to be revised. Not all amplitude modulated whistles have sidebands but only those that actually contain pulses. You can have an amplitude modulated continuous wave without sidebands but harmonics.

Line 93-94: I don’t think you can throw the ultrasonic whistles/high-frequency modulated signals into the same description as relatively low-frequency “standard” whistles. That slipped my notice last time around. These are two distinctly different categories and very separate from each other in frequencies as well as likely behavioral use.

Line 108-109: It seems that the term amplitude-modulation is used interchangeably with pulse-repetition rate. That doesn’t seem appropriate here or in lines 91-92.

Line 167-168: What was the time window? And what was the size of the window on your screen? Ultimately, if you manually pick your measurements then your time/frequency precision is only partially limited by your FFT and overlap but also by the number of pixels available on your screen to display that resolution.

Line 365: “comprehensive” – this is a strong statement; it’s a large repertoire but may not be comprehensive (as indicated in discussion).

Figure 1: Map labels (axes and other text) too small.

Figures 2-8: the actual frequency and time values are still way too small, especially in figure 8.

Figure 8: Comparative graphs should be scaled in time and frequency such that they are actually comparable and legible.

Appendix 2: Table 1 – would it be possible, maybe instead of column bit depth, which doesn’t change and isn’t too important anyways, to include recording duration?

Decision letter (RSOS-191228.R1)

19-Dec-2019

Dear Ms Wellard,

On behalf of the Editors, I am pleased to inform you that your Manuscript RSOS-191228.R1 entitled "Cold Call: The Acoustic Repertoire of Ross Sea Killer Whales (*Orcinus orca*, Type C) in McMurdo Sound, Antarctica" has been accepted for publication in Royal Society Open Science subject to minor revision in accordance with the referee suggestions. Please find the referees' comments at the end of this email.

The reviewers and Subject Editor have recommended publication, but also suggest some minor revisions to your manuscript. Therefore, I invite you to respond to the comments and revise your manuscript.

- Ethics statement

- Data accessibility

It is a condition of publication that all supporting data are made available either as supplementary information or preferably in a suitable permanent repository. The data accessibility section should state where the article's supporting data can be accessed. This section should also include details, where possible of where to access other relevant research materials

such as statistical tools, protocols, software etc can be accessed. If the data has been deposited in an external repository this section should list the database, accession number and link to the DOI for all data from the article that has been made publicly available. Data sets that have been deposited in an external repository and have a DOI should also be appropriately cited in the manuscript and included in the reference list.

If you wish to submit your supporting data or code to Dryad (<http://datadryad.org/>), or modify your current submission to dryad, please use the following link:
<http://datadryad.org/submit?journalID=RSOS&manu=RSOS-191228.R1>

- **Competing interests**

- **Authors' contributions**

- **Acknowledgements**

- **Funding statement**

Because the schedule for publication is very tight, it is a condition of publication that you submit the revised version of your manuscript before 28-Dec-2019. Please note that the revision deadline will expire at 00.00am on this date. If you do not think you will be able to meet this date please let me know immediately.

When submitting your revised manuscript, you will be able to respond to the comments made by the referees and upload a file "Response to Referees" in "Section 6 - File Upload". You can use this

to document any changes you make to the original manuscript. In order to expedite the processing of the revised manuscript, please be as specific as possible in your response to the referees.

on behalf of Dr Ari Friedlaender (Associate Editor) and Kevin Padian (Subject Editor)
openscience@royalsociety.org

Associate Editor Comments to Author (Dr Ari Friedlaender):
To the Authors,

The two reviewers and I agree that you have been diligent with your revisions and that your manuscript is suitable for publication pending a few minor comments that require attention. I am pleased with how you have treated the data, the comments from the reviewers, and crafted a very nice piece of work that will be of interest to a broad readership. I congratulate you on the fine effort and look forward to seeing this published.

Thank you.
Ari S. Friedlaender

Reviewer comments to Author:

Reviewer: 1

Comments to the Author(s)

In this revision, Wellard et al. have made substantial effort to address reviewer comments and edits. I have only a few very minor additional comments, and think that the study is suitable for publication.

Introduction

Lines 106-110 – This could possibly be restructured a bit, the way it currently reads sounds like graded call are a phenomenon unique to killer whales and false killer whales, but it's likely that it occurs in many of the odontocetes, we just haven't done the research yet to test that. For example, there is some evidence of graded call structures in wild populations of pilot whales.

Methods

Lines 157-158 – note the number of encounters/recordings for each of the sampling rates. Based on your explanation, it sounds like n=1 encounter at 44 kHz sampling rate and n=8 encounters at 96 kHz, is that correct? Add a sentence here explaining how you controlled for differences in the sampling rate, or explaining why that was not necessary.

Discussion

367 – 167 individuals were reported in the results, should be changed here to match.

Reviewer: 2

Comments to the Author(s)

This version of a manuscript describing the vocal repertoire of Antarctic Type C killer whales has been much improved. However, I still see a few slight necessary changes, see details below. Particularly the legibility of the figure values is of concern, even if the figures are provided with 300 dpi.

Line 52: First figure mention – numbering should start with “1” here, suggest changing the order.

Line 91-92: The reference “42” talks about pulsed calls, not amplitude-modulated whistles. This needs to be revised. Not all amplitude modulated whistles have sidebands but only those that actually contain pulses. You can have an amplitude modulated continuous wave without sidebands but harmonics.

Line 93-94: I don't think you can throw the ultrasonic whistles/high-frequency modulated signals into the same description as relatively low-frequency “standard” whistles. That slipped my notice last time around. These are two distinctly different categories and very separate from each other in frequencies as well as likely behavioral use.

Line 108-109: It seems that the term amplitude-modulation is used interchangeably with pulse-repetition rate. That doesn't seem appropriate here or in lines 91-92.

Line 167-168: What was the time window? And what was the size of the window on your screen? Ultimately, if you manually pick your measurements then your time/frequency precision is only partially limited by your FFT and overlap but also by the number of pixels available on your screen to display that resolution.

Line 365: “comprehensive” – this is a strong statement; it's a large repertoire but may not be comprehensive (as indicated in discussion).

Figure 1: Map labels (axes and other text) too small.

Figures 2-8: the actual frequency and time values are still way too small, especially in figure 8.

Figure 8: Comparative graphs should be scaled in time and frequency such that they are actually comparable and legible.

Appendix 2: Table 1 – would it be possible, maybe instead of column bit depth, which doesn't change and isn't too important anyways, to include recording duration?

Author's Response to Decision Letter for (RSOS-191228.R1)

See Appendix D.

Decision letter (RSOS-191228.R2)

09-Jan-2020

Dear Ms Wellard,

It is a pleasure to accept your manuscript entitled "Cold Call: The Acoustic Repertoire of Ross Sea Killer Whales (*Orcinus orca*, Type C) in McMurdo Sound, Antarctica" in its current form for publication in Royal Society Open Science.

Kind regards,
Lianne Parkhouse
Editorial Coordinator
Royal Society Open Science
openscience@royalsociety.org

on behalf of Dr Ari Friedlaender (Associate Editor) and Kevin Padian (Subject Editor)
openscience@royalsociety.org

Appendix A

Wellard et al. describe the vocal repertoire of Type C killer whales found in the Ross Sea MPA, which can be used for acoustic monitoring of this top predator and ecosystem indicator, in a region that is remote and otherwise difficult to study. The study builds on previous work to document killer whale vocalizations in the Ross Sea, and improves upon this with a larger sample size (number of encounters and number of individuals recorded). Additionally, photos were taken during the encounters to identify group ecotype. The authors identified 29 distinct call types from their dataset. The call catalogue generated for this study, when compared with call catalogues from previous studies, indicates that at least some call types are temporally stable. The authors suggest that this work, in addition to its usefulness in acoustic monitoring, may be used to understand variability among Antarctic ecotypes, and may be combined with genetic data to understand patterns of cultural information transfer (e.g. matrilineal, vertical, horizontal).

Overall, the study seems well designed and complete, and provides important baseline information to further study killer whale ecology and evolution in the Antarctic. I have some concerns, outlined below, which should either be justified in the manuscript or revisited.

Abstract

How many different encounters and unique individuals in the 3hrs33min?

Introduction

Line 33 – include species name at first mention of killer whale

Line 47 – include average or maximum adult male size for the species, for comparison

Line 65 – instead of “that detect”, “used to detect”

Lines 83-96 – You include characteristics of whistles from other killer whale populations, but no characteristics of clicks or burst pulses. Why not? The info would be useful for comparison.

Methods

2.1 Data collection

Line 142 – additional specifications for the hydrophone and acoustic recorder are necessary, especially for the custom-built device, such as flat response range (or functional bandwidth), pre-amp flat response range, recorder bit-depth/resolution. Also, please be specific about when each sampling frequency was used. I’m concerned about the 44.1 kHz sampling rate used for some recordings – 22 kHz is far below the upper bounds of killer whale vocalizations, so these recordings may be biasing your results toward lower frequency calls. Please justify the use of these low-frequency recordings.

2.3 Call type categorization

Line 163 – please describe how you defined a “component” for the purpose of this analysis.

Line 167 – I’ve seen studies classify call types if they are repeated 10, 5, or even 3 times. Twice seems low. If a single animal produces a sound two times and then never again, should it be considered a call type? Will categorizing those calls be useful for monitoring or follow-up analyses?

Line 196 – if measurements were calculated for each component separately, how were multi-component calls handled? For example, did you add the duration of each individual component to get the overall duration of a multi-component call? For maximum frequency, did you take the maximum frequency from all the components in that call?

Lines 199-202 – This is confusing and needs re-wording; might be easier to explain with an example figure.

Results

Line 218 – based on photos collected during these ecounters, can you confirm that the 9 encounters were with different individuals or groups of animals? How many encounters were with different groups vs. repeat encounters?

Line 236 – Table 2 seems to be missing? Or is this referring to the table in the appendix? The wording is confusing.

Interobserver reliability – please define and provide context for the values reported in the Fleiss-Kappa statistic. It might be worthwhile to supplement with a more readily interpretable number, such as a confusion matrix between the naïve observer classification and the call catalogue (the call catalogue taken as “truth” in the confusion matrix). This would also highlight whether there is a bias toward confusion in a few calls, or if confusion is spread generally amongst all calls in the study.

Lines 246-248 – Given that some calls were found over 100 times in the dataset, I’m again drawn to question why calls that were repeated twice were included as call types. How often were the other 22 call types repeated? How many were repeated fewer than 5 or 10 times, and is there a reason to include these? It might be worthwhile to include a bar plot showing how many times each call was repeated.

Line 252 – I think here you are referring to Figure 3.

Line 262 – “21 out of 29 call categories contained multi-component calls...” I am curious what this statistic looks like if you include calls that were repeated more than 5 or 10 times, rather than just twice.

Discussion

Line 320 – “focused”

Line 327 – The number of call types identified in each study is also likely to be strongly affected by sample size, and the number/types of encounters included in the study, since animals will produce different calls during different times of year, in different locations, or different behavioral states. An additional caveat is the number of times a call needs to be repeated before being considered a call type – since two is a particularly low number, it make sense that this study produced more call types than other studies with a (presumably) higher threshold for including a call type in the study.

Line 339 – how did you measure the calling rate in your study? and how did you quantify acoustic variability?

Line 359 – “A biphonation appears...” the definition of a biphonation should be included the first time the term is mentioned, rather than in the discussion.

Line 375 – “having” should be “have”. Also, do you mean up to 50, or over 50? Up to 50 could be anywhere from 1-50, but here it seems like you’re talking about larger group sizes.

Line 418 – what seven calls are you referring to here?

Lines 425-431 – What about the 66% of Richlen calls that did not match your catalogue? Or the Billon calls that did not match your catalogue? Does this suggest that the vocal repertoire is still only partially described, and that further sampling needs to be undertaken to fully describe the repertoire? Or do you think there is long-term temporal shifting in the vocal repertoire of these animals, and those historically recorded calls have been permanently lost? What percentage of the Richlen and Billon datasets are from calls that were not found in your study?

Line 453 – per Conner (1982), the term “dialect” should only be used to describe differences in the vocal repertoires of sympatric populations within a species. Are these killer whales sympatric with other populations, and if so, can you show that their vocal repertoire is different from those populations?

Line 457 – Region should probably be capitalized.

Line 480-481 – Finding a large number of call types is not, in and of itself, an indication that you have captured most of the repertoire. You might try a rarefaction curve (normally used for species richness or to understand how well the species in a region are sampled) if you would like to get an idea of how completely you have sampled the vocal repertoire of the population, but keep in mind the underlying assumptions that all calls occur with roughly the same frequency, and that the distribution of calls will be random with respect to the sampling scheme.

Line 480 – Please clarify “high rate of encounter of individuals” – does this mean you encountered the same individuals many times? Or that you encountered a large portion of the population? I didn’t see any estimates of the Type C killer whale population size, so I don’t know what proportion of the population is represented by the estimated 353 individuals that were encountered (and of those individuals, how many were repeat-sampled vs. how many were unique?). It would be helpful to include this information in the results to support this statement.

Line 494 – I suggest replacing “dialect” with “vocal repertoire”. As with above, “dialect” is a technical term used to refer to variability in the vocal repertoires of sympatric populations (Conner 1982).

Figures

Please make axis labels and axis titles larger for all figures.

Figure 3: It would be helpful to point out or somehow highlight the component in panel (a) that is missing in panel (b), likewise for panels (c) and (d).

Figure 4: The figure caption is long and repetitive, especially since you say in the first line that all of the calls are multi-component. I suggest removing the description of each panel separately and just say that panel (a) has a biphonation and panel (b) has two components.

Table 1: description of “Duration 90%” is confusing, needs re-wording. Can use similar wording to “Bandwidth 90%”, e.g. “the duration (s) containing 90% of the call energy”

Appendix B

Specific comments:

Introduction:

Comment 1

Line 43: The switch between “Ross Sea killer whales” and “Type C killer whales” is confusing. It seems like Type C would make more sense so I suggest changing to “Antarctic Type C killer whales” throughout the paper.

Comment 2

Line 65: Should be “detects”.

Comment 3

Line 116: Should be “Ice Shelf”.

Comment 4

Line 128: PAM was not defined previously and there is a lot of switching between “PAM” and “passive acoustic monitoring”. Please make this consistent throughout the paper.

Methods:

Comment 5

Lines 139-140: What depth was the hydrophone deployed at while recording? What was the water depth?

Comment 6

Lines 140-142: Are you able to provide any more information about your recording setup? Was an amplifier or filter used?

Comment 7

Line 144: Were recordings started before you had confirmation that you were observing Type C killer whales? Were other Antarctic types ever recorded? Is it possible you recorded multiple types in one recording?

Comment 8

Line 156-158: How many calls were Grade 2 vs. Grade 3? Since you found such a large number of discrete call types, I’m curious how the call type categories would change if only Grade 3 calls were used.

Comment 9

Line 163-165: It might be helpful here to include a figure of an example call type where you indicate the different features you used for classification (i.e., showing what biphonation looks like, how the call breaks down into different components, etc.). I would recommend including this sort of figure instead of Table 1 (see Comment 26 on Table 1 below).

Comment 10

Line 167: Is there support in the literature for keeping a call type with only 2 examples? With only 2 examples I would worry that some of the call types are actually aberrant calls. Sharpe et al., 2017 required three or more examples to try and avoid this problem.

Comment 11

Lines 172-174: This sentence describing biphonic vs. monophonic seems out of place. I suggest moving it to the preceding paragraph.

Comment 12

Lines 186-190: How did you determine which parameters to measure? How did you decide which were “useful” for burst-pulse sounds vs. those that were “useful” for whistles?

Comment 13

Line 204: I would clarify what you mean by “other killer whales” here. I assume you mean other Type C killer whales in the same group, but it could be construed to mean a different Antarctic Type.

Results:

Comment 14

Line 218: It would be good to give some more information about each of the nine encounters that you recorded. Perhaps you could include a table with the day and time, number of individuals, behaviors noted, and call types recorded.

Comment 15

Lines 233-235: Please verify the total number of calls given here. Summing the call totals given in the Supplementary Material results in 1253 calls, which is larger than the total given in the text before some calls were removed. Therefore, the number reported in the text should be higher, or it needs rewording for clarity.

Comment 16

Lines 234-235: 29 categories (including 4 subtypes) matches what is shown in the call catalogue (Supplementary Material, Appendix 2) but it does not match the information in Table 2 (Supplementary Material, Appendix 1). The Table includes a 5th subtype (McM10a) that is not in the call catalogue. This needs to be changed or explained somewhere.

Comment 17

Lines 236-237: What is meant by “with further parameters measured” here? Aren’t these the same parameters given in Table 2 (Appendix 1)?

Comment 18

Line 246: Call types are referred to only by number here and not with a preceding “McM” as is done later in the text. Please make this consistent.

Comment 19

Would it be better to exclude the “McM” from the call types defined here when there has been a previous publication with confirmed Type C killer whale calls? It is confusing when every publication chooses their own naming scheme for call types. Perhaps this should be adding on to the findings of Schall and Van Opzeeland, 2017.

Comment 20

Line 252: Should reference Figure 3.

Comment 21

Line 263: Should reference Figure 4.

Comment 22

Line 273: Should reference Figure 5.

Discussion:

Comment 23

Lines 355-357: Is there any information about this acoustic feature in other ecotypes in different areas to suggest it is unique to Antarctic killer whales and might be useful for identification?

Comment 24

Lines 368-369: If you want to make claims like this you need to show your data. As mentioned previously, it would be helpful to see a breakdown of the number of individuals present and behaviors documented during the encounters where recordings were made.

References:

Comment 25

Formatting of references needed. The journal names are only abbreviated for some of the references.

Tables and Figures:

Comment 26

Table 1: This table is unnecessary in the text and would be better suited for only the supplementary material. It seems strange to include such a lengthy table with the definitions for the parameters when none of the actual measurements are provided in the main paper. Additionally, it seems misleading to present so much information about these parameters in the methods when they were not actually used in constructing the call catalogue.

Comment 27

Figure 6: Ideally the spectrograms in this figure could be larger as it isn't possible to read the axes labels without zooming in on the pdf. Making the axes labels clear is particularly important here since they vary between sources. Would it be possible to add larger axes labels separately?

Comment 28

The axes for all the figures are too small to read without zooming in, as are the location labels in Figure 1. The figures are high enough resolution that you can zoom in to read these but having to zoom in on everything is not ideal. It would be better to make the axes all larger if possible.

Supplementary Material:

Comment 29

In Table 2, the call types are listed as numbers instead of being preceded by “McM” as they are in the text and Appendix 2. The naming scheme should be consistent throughout.

Comment 30

In Table 2, the column heading “Component” is used twice. Perhaps “Component” and then “Component Type”, where the first column has the component number and the second notes whether the component was a whistle, burst-pulse, or biphonic whistle.

Comment 31

In Table 2, the n given for each call type is the total number of calls you classified for that category. This is misleading because you state in the text that you didn’t necessarily measure all of the calls for each category. Since you are presenting your parameter measurements in this table you should instead be reporting the number of calls you actually measured, especially since you don’t give exact numbers in the text for this.

Comment 32

The note for Table 2 should read “has measurements”.

Comment 33

It would make more sense to include Table 2 at the end of the call catalogue in Appendix 2. Currently, the table is repeated in both Appendices but in Appendix 2 it is broken up and does not have a legend.

Comment 34

In Appendix 2, the description of McM1 should read “This is a 4-component and biphonic call” to be consistent with subsequent call descriptions.

Comment 35

In Appendix 2, the description of McM10 mentions this call being produced in a “rhythmic repeated call sequence”. Could you elaborate on what is meant by this? Is this related to there being a call type McM10a in Table 2 but not in the catalogue? What percentage of the time was this call produced in a sequence?

Appendix C

Dear Editors of Royal Society Open Science,

This letter is in reference to our submitted manuscript to your journal Royal Society Open Science:

Manuscript ID: RSOS-191228

Title: Cold Call: The Acoustic Repertoire of Ross Sea Killer Whales (*Orcinus orca*, Type C) in McMurdo Sound, Antarctica.

Please see attached our Response to Referees.

Below are responses to each point brought up by the reviewers.

Reviewer #1 Comments and Responses

REVIEWER COMMENT	RESPONSE
Abstract How many different encounters and unique individuals in the 3hrs33min?	The abstract states there are 9 different encounters for the period of 3hrs and 33 mins of acoustic recordings. Full details of identified individual killer whales are now given in the Results section- this is explained in depth and would not fit in the abstract. For full transparency, there is also a table in the Supplementary Material detailing every killer whale encounter during the study, along with other information such as behaviours observed, visual estimates, number of identified killer whales per encounter and call categories recorded during each encounter.
Introduction Line 33 – include species name at first mention of killer whale	Have added in species name at first mention of killer whale in Introduction.
Line 47 – include average or maximum adult male size for the species, for comparison	Have added in maximum length of Type A killer whale for comparison in line 47: “...adult Type C males up to 6.1 m, in comparison to Type A males up to 9.2 m;...”
Line 65 – instead of “that detect”, “used to detect”	Have edited as requested.
Lines 83-96 – You include characteristics of whistles from other killer whale populations, but no characteristics of clicks or burst pulses. Why not? The info would be useful for comparison.	I have now edited this section to include more details on the structure of burst pulse sounds and described characteristics of killer whale burst pulse sounds from other killer whale populations worldwide. Only one study has ever discussed population-level difference in killer whale echolocation clicks (Barrett-Lennard et al., 1996), which is why I only compare characteristics of whistles and bps in killer whale populations.

Methods 2.1 Data collection Line 142 – additional specifications for the hydrophone and acoustic recorder are necessary, especially for the custom-built device, such as flat response range (or functional bandwidth), pre-amp flat response range, recorder bit-depth/resolution. Also, please be specific about when each sampling frequency was used. I’m concerned about the 44.1 kHz sampling rate used for some recordings – 22 kHz is far below the upper bounds of killer whale vocalizations, so these recordings may be biasing your results toward lower frequency calls. Please justify the use of these low-frequency recordings.	Additional specifications for the hydrophone have now been included in the manuscript. See in Methods. The sampling frequency was inadvertently switched from 96 kHz to 44.1 kHz on the last day of the study. This was the only day at which recordings had a sampling frequency of 44.1 kHz and 24-bit, all other recordings sampled at a rate of 96 kHz and 24-bit. A full detailed list of every recording and the sampling and bit depth is now included in Table 1 in the Supp Material, Appendix 2, for full transparency.
2.3 Call type categorization Line 163 – please describe how you defined a “component” for the purpose of this analysis.	This is described in lines 224-233, which I have now expanded on to include how components were defined. It now reads: “In order to describe the different components of the recorded calls, we used the default from Watkins (1968) calling sounds with fewer than five harmonics a whistle, and those with more contours a burst-pulse sound. We also note that the majority of calls recorded transitioned gradually from burst-pulses to whistles and vice versa by increasing the pulse-repetition rate and decreasing the inter-pulse interval until continuous tones were formed, and vice versa. This category of transition calls follows Murray et al. and their definition of characterizing graded vocalisations and the continuum from whistle to pulses. Calls were divided into components not only when they transitioned from a whistle to burst-pulse sound, or vice versa, but also when there were rapid shifts in the vocalisation’s pulse repetition rate, as per Yurk.”
Line 167 – I’ve seen studies classify call types if they are repeated 10, 5, or even 3 times. Twice seems low. If a single animal produces a sound two times and then never again, should it be considered a call type? Will categorizing those calls be useful for monitoring or follow-up analyses?	I agree with this comment and have now excluded call groups that had calls repeated twice. I have set a new minimum of 3 calls required to define a call type, as per other studies e.g. Sharpe et al. This is reflected in the new call catalogue, new numbering system and the total number of call types in this manuscript now.
Line 196 – if measurements were calculated for each component separately, how were multi-component calls handled? For example, did you add the duration of each individual component to get the overall duration of a multi-component call? For maximum frequency, did you take the maximum frequency from all the components in that call?	This is answered in line 226 (“For multi-component calls, parameters were measured separately for each component”) and detailed for every call category in the Appendix Call Catalogue. All parameters in Table 1 were measured separately for all components of a call. Along with every component measured separately, the parameters of the entire call was also measured. The results of these measurements are detailed in the call catalogue for each call category.

Lines 199-202 – This is confusing and needs re-wording; might be easier to explain with an example figure.	I have edited these lines, so they are less confusing. They now stand at: “Frequency measurements such as start, end, minimum, and maximum frequency are scaled; i.e., a factor n higher for the n^{th} harmonic (where the fundamental was counted as the 1st harmonic and the first overtone was considered the 2nd harmonic). For example, if measurements were done off the first overtone, then the measurements were divided by 2 in order to correspond to the fundamental.”
Results Line 218 – based on photos collected during these encounters, can you confirm that the 9 encounters were with different individuals or groups of animals? How many encounters were with different groups vs. repeat encounters?	We have now undertaken photo-identification analysis for this study. All photos for every encounter have been processed and the results now include the number of identified individual killer whales per encounter. This is in the results section in the manuscript, and also in greater detail in Table 1 in the Supplementary Material, Appendix 2, for full transparency. In Results section: “A total of 167 individuals were identified in the nine encounters, based on the published photo-ID catalogue. The maximum number of resights of the same animal was 4 (i.e., this animal was seen in four out of the nine encounters) on three different days. About 392 killer whales in total were estimated over the nine encounters, although some of these individuals were likely re-sights, while subsurface individuals might have been missed during visual counting. “
Line 236 – Table 2 seems to be missing? Or is this referring to the table in the appendix? The wording is confusing.	Table 2 is in the Supplementary as stated in the sentence. “Summary statistics for the acoustic parameters of each call type are listed in Table 2 (Supplementary Material, Appendix 1)...”. I am happy to change this if the Editor of RSOS would like me to reference supplementary material differently.
Interobserver reliability – please define and provide context for the values reported in the Fleiss-Kappa statistic. It might be worthwhile to supplement with a more readily interpretable number, such as a confusion matrix between the naïve observer classification and the call catalogue (the call catalogue taken as “truth” in the confusion matrix). This would also highlight whether there is a bias toward confusion in a few calls, or if confusion is spread generally amongst all calls in the study.	Kappa is a measure of difference, standardized to lie on a -1 to 1 scale, where 1 is perfect agreement, 0 is exactly what would be expected by chance, and negative values indicate agreement less than chance, ie, potential systematic disagreement between the observers. Therefore, if the observers who undertook the test for interobserver reliability on these Type C call categories were in complete agreement in the classification, then the Fleiss’ Kappa statistic (κ) would be equal to 1. If the agreement amongst judges is the same as would be expected by chance, then κ would be equal to 0. With a κ value of 0.515, this is considered moderate agreement according to Landis and Koch (1977). I have now added in this information and placed the results of the Kappa statistic in context to what the test revealed for our analysis. A confusion matrix is beyond the scope of this interobserver reliability test and not necessary to be displayed for readers to understand this analysis.

Lines 246-248 – Given that some calls were found over 100 times in the dataset, I’m again drawn to question why calls that were repeated twice were included as call types. How often were the other 22 call types repeated? How many were repeated fewer than 5 or 10 times, and is there a reason to include these? It might be worthwhile to include a bar plot showing how many times each call was repeated.	I have now changed this requirement for categorisation at per request of this reviewer. I have excluded call groups that had calls repeated twice. I have set a new minimum of 3 calls required to define a call type, as per other studies e.g. Sharpe et al. This is reflected in the new call catalogue, new numbering system and the total number of call types in this manuscript now. I have also included a bar graph showing how many times each call was repeated at the request of this reviewer and added this graph to the Call Catalogue.
Line 252 – I think here you are referring to Figure 3.	Correct. Have changed this accordingly.
Line 262 – “21 out of 29 call categories contained multi-component calls...” I am curious what this statistic looks like if you include calls that were repeated more than 5 or 10 times, rather than just twice.	The number of calls per call type are all provided in the Supplementary Material, along with information on multiple component vs. single component call types.
Discussion Line 320 – “focused”	Correct. Have changed this accordingly.
Line 327 – The number of call types identified in each study is also likely to be strongly affected by sample size, and the number/types of encounters included in the study, since animals will produce different calls during different times of year, in different locations, or different behavioral states. An additional caveat is the number of times a call needs to be repeated before being considered a call type – since two is a particularly low number, it make sense that this study produced more call types than other studies with a (presumably) higher threshold for including a call type in the study.	I agree with this comment. The difference in call rate and call types can be impacted by the number of encounters/individuals (sample size) and behavioural state. This is discussed in lines 329-345. I have also now adjusted a requirement of a minimum of 3 calls recorded to delineate a call type.
Line 339 – how did you measure the calling rate in your study? and how did you quantify acoustic variability?	I have removed all references to the high calling rate in our study, as this would require further explanation of analysis and is beyond the scope of this paper designed to purely describe the call repertoire. Acoustic variability is in relation to the number of call types and variation in structural components found in the Type C killer whale call repertoire, in comparison other killer whale repertoires described worldwide. This is also discussed in lines 346-348.
Line 359 – “A biphonation appears...” the definition of a biphonation should be included the first time the term is mentioned, rather than in the discussion.	This definition is given early on in the manuscript, in the methods section, line 176: “Calls were classified as biphonic if they had two simultaneous but independently modulated frequency components, otherwise they were classified as monophonic”.
Line 375 – “having” should be “have”. Also, do you mean up to 50, or over 50? Up to 50 could be anywhere from 1-50, but here it seems like you’re talking about larger group sizes.	Have changed to “have”. Also changed this wording regarding group sizes, as it was previously confusing. All group sizes and

	encounter data are given in Table 3 in Appendix 2, for full transparency.
Line 418 – what seven calls are you referring to here?	This is referring to the results from the call comparison, see lines 308-328: section 3.2 “Call comparison with killer whale call repertoires described elsewhere in the Southern Hemisphere”, Figure 6. I can refer to Figure 6 here if the editors would like to clarify, but it is uncommon to refer to Figures in discussion.
Lines 425-431 – What about the 66% of Richlen calls that did not match your catalogue? Or the Billon calls that did not match your catalogue? Does this suggest that the vocal repertoire is still only partially described, and that further sampling needs to be undertaken to fully describe the repertoire? Or do you think there is long-term temporal shifting in the vocal repertoire of these animals, and those historically recorded calls have been permanently lost? What percentage of the Richlen and Billon datasets are from calls that were not found in your study?	Some good questions posed by this reviewer. Call stability and long-term temporal shifting in vocal repertoire is addressed in section 4.5: Call stability in the Ross Sea killer whale repertoire. As this is the first study to describe the vocal repertoire with confirmed ecotype Type C in McMurdo Sound, further sampling does need to be done to fully describe the repertoire. This is discussed further in the manuscript in lines 522-529, “This study catalogued 28 complex and clearly recognisable call types of Type C killer whales and while the catalogue may not be complete, the number of call types and subtypes, along with the large number of individuals encountered, suggests a good portion of the repertoire may have been captured. Future research should investigate the call repertoire of other Antarctic killer whale ecotypes and examine the potential acoustic divergence.”
Line 453 – per Conner (1982), the term “dialect” should only be used to describe differences in the vocal repertoires of sympatric populations within a species. Are these killer whales sympatric with other populations, and if so, can you show that their vocal repertoire is different from those populations?	Yes, I completely agree with this comment. Here we discuss this definition of a dialect in sympatric populations and geographical variation. Type C killer whales live sympatrically with Type B killer whales in the Ross Sea. In line 502 we only hypothesize that Type C killer whales may have a distinct dialect, but further comparative acoustic research is needed.
Line 457 – Region should probably be capitalized.	Have changed to capitalized.
Line 480-481 – Finding a large number of call types is not, in and of itself, an indication that you have captured most of the repertoire. You might try a rarefaction curve (normally used for species richness or to understand how well the species in a region are sampled) if you would like to get an idea of how completely you have sampled the vocal repertoire of the population, but keep in mind the underlying assumptions that all calls occur with roughly the same frequency, and that the distribution of calls will be random with respect to the sampling scheme.	Some good advice. I have reworded these lines so it no longer states that a large portion of the repertoire may have been captured. We also state in the introduction that this study is an initial step towards distinguishing the Type C killer whale repertoire. A rarefaction curve is beyond the scope of this paper.
Line 480 – Please clarify “high rate of encounter of individuals” – does this mean you encountered the same individuals many times? Or that you encountered a large portion of the population? I	We have now undertaken photo-identification analysis for this study. All photos for every encounter have been processed and the results now include the number of identified individual killer whales per

didn't see any estimates of the Type C killer whale population size, so I don't know what proportion of the population is represented by the estimated 353 individuals that were encountered (and of those individuals, how many were repeat-sampled vs. how many were unique?). It would be helpful to include this information in the results to support this statement	encounter. We have also added relevant information into the manuscript, such as the population estimate reported by a photo-identification study from Pitman et al. (2018). In total, there were 167 different individuals identified across the sightings when recordings were made. This is in the results section in the manuscript, and also in greater detail in Table 1 in the Supplementary Material, Appendix 2, for full transparency. We have removed any statements regarding high rate of encounter of individuals.
Line 494 – I suggest replacing “dialect” with “vocal repertoire”. As with above, “dialect” is a technical term used to refer to variability in the vocal repertoires of sympatric populations (Conner 1982).	Have adjusted to “vocal repertoire” upon request.
Figures Please make axis labels and axis titles larger for all figures.	All axis labels and spectrograms have been redone with larger font and titles for all figures. I also have done these spectrograms in grey scale if the editors would prefer this colour grade rather than the blue colour grading.
Figure 3: It would be helpful to point out or somehow highlight the component in panel (a) that is missing in panel (b), likewise for panels (c) and (d).	This figure has now been edited to highlight the component that is present in the primary call, but absent from the subtype.
Figure 4: The figure caption is long and repetitive, especially since you say in the first line that all of the calls are multi-component. I suggest removing the description of each panel separately and just say that panel (a) has a biphonation and panel (b) has two components.	Have reworded figure caption as per suggestion- individual descriptions of multi-component calls have been removed.
Table 1: description of “Duration 90%” is confusing, needs re-wording. Can use similar wording to “Bandwidth 90%”, e.g. “the duration (s) containing 90% of the call energy”	Have changed description of Duration 90% in table. Now reads: Useful for burst-pulse sounds and whistles, the duration [s] containing 90% of the call energy.

Reviewer #2 Comments and Responses

REVIEWER COMMENT	RESPONSE
Introduction: Comment 1 Line 43: The switch between “Ross Sea killer whales” and “Type C killer whales” is confusing. It seems like Type C would make more sense so I suggest changing to “Antarctic Type C killer whales” throughout the paper.	We have changed now changed this throughout the paper now. The title will stay the same (unless further review would like it changed). Currently the manuscript only references the RSKW on line 45, otherwise all references in the manuscript state Type C. Line 45: “Type C killer whales (also known as Ross Sea killer whales; RSKW,) are known ...”
Comment 2 Line 65: Should be “detects”.	Have edited line 65 to: “...passive acoustic monitoring is a technique used to detect occurrence and relative abundance in the longer term”
Comment 3 Line 116: Should be “Ice Shelf”.	Have edited to “Ice Shelf”.
Comment 4 Line 128: PAM was not defined previously and there is a lot of switching between “PAM” and “passive acoustic monitoring”. Please make this consistent throughout the paper.	Have now introduced passive acoustic monitoring (PAM) in line 67, and from here on in, will use abbreviation PAM.
Methods: Comment 5 Lines 139-140: What depth was the hydrophone deployed at while recording? What was the water depth?	This has been answered in the Methods: “The water depth in McMurdo Sound ranges from a shallow 200-400 m slope on the west side to a 700 m deep trench on the east side.”; and also: “The hydrophone was hand-deployed into the water at the ice edge in the immediate vicinity of killer whales (i.e., at < 100 m range) approximately 3-4 m below the surface.”
Comment 6 Lines 140-142: Are you able to provide any more information about your recording setup? Was an amplifier or filter used?	Additional specifications for the recording setup have now been included in the manuscript in the Methods section.
Comment 7 Line 144: Were recordings started before you had confirmation that you were observing Type C killer whales? Were other Antarctic types ever recorded? Is it possible you recorded multiple types in one recording?	Recordings commenced once killer whales were detected and ecotype identified (line 135-146). One other ecotype was observed (Type B), but all files containing another ecotype present were excluded- this is reported on line 153: “Only recordings made during a confirmed sighting of Type C killer whales were included in our analysis.”
Comment 8 Line 156-158: How many calls were Grade 2 vs. Grade 3? Since you found such a large number of discrete call types, I’m curious how the call type categories would change if only Grade 3 calls were used.	Grade 2 calls are suitable for acoustic analysis with a distinct and clear signal, allowing all required parameters to be measured. Hence, I’m not sure on the relevance of this question. A lot of calls were excluded from the analysis (Grade 1) due to poor SNR and a faint signal. There is no reason Grade 2 calls should be excluded. A total of 6386 killer whale vocalisations were initially detected and subsequently graded. A total of 5134 Grade 1 calls were removed. Categories with fewer than three examples of each type were also removed yielding a total of 1250

	vocalisations placed into 28 categories, inclusive of 4 subtypes. This is detailed in the manuscript and also now in the Call Catalogue.
Comment 9 Line 163-165: It might be helpful here to include a figure of an example call type where you indicate the different features you used for classification (i.e., showing what biphonation looks like, how the call breaks down into different components, etc.). I would recommend including this sort of figure instead of Table 1 (see Comment 26 on Table 1 below).	Great advice. There are now 2 new figures added to meet these questions. The first new image illustrates how a call is broken into segments/components, and what a biphonation looks like. The second figure illustrates the parameters measured for acoustic analysis. Both these figures have replaced Table 1 in the manuscript, and Table 1 has now been moved to Appendix 1 in the Supplementary Material.
Comment 10 Line 167: Is there support in the literature for keeping a call type with only 2 examples? With only 2 examples I would worry that some of the call types are actually aberrant calls. Sharpe et al., 2017 required three or more examples to try and avoid this problem.	Good advice, and similar to comments from Reviewer 1. I have now adjusted a requirement of a minimum of 3 calls recorded to delineate a call type. Similar to Sharpe et al. 2017 standards.
Comment 11 Lines 172-174: This sentence describing biophonic vs. monophonic seems out of place. I suggest moving it to the preceding paragraph.	Have moved lines 172-174 to preceding paragraph, now at line 164-165.
Comment 12 Lines 186-190: How did you determine which parameters to measure? How did you decide which were “useful” for burst-pulse sounds vs. those that were “useful” for whistles?	To quantify the spectro-temporal structure of each call, we selected the most appropriate parameters available in the Raven Pro package suitable for the type of vocalisation: burst-pulse sounds that are broadband calls vs. whistles that are frequency modulated calls. This is described in lines 186-189. It currently stands at: “Some of the parameters are more useful for quantifying broadband calls like burst-pulse sounds (e.g., entropy measures and quartile frequencies), while others are more useful for whistles (e.g., start, end, minimum and maximum frequencies of the contour).
Comment 13 Line 204: I would clarify what you mean by “other killer whales” here. I assume you mean other Type C killer whales in the same group, but it could be construed to mean a different Antarctic Type.	Yes, I mean other killer whales in the same group. Sorry for any confusion. I will add that into the sentence for clarification. The sentence now stands at: “...due to ice noise, overlapping sounds from other killer whales in the group and recording artefacts”.
Results: Comment 14 Line 218: It would be good to give some more information about each of the nine encounters that you recorded. Perhaps you could include a table with the day and time, number of individuals, behaviors noted, and call types recorded.	Good suggestion. A table has now been included in the Call Catalogue, Appendix 2, detailing all this information, as requested.
Comment 15	Here in the results it is stated there are a total of 1252 vocalisations, whilst in the supp material it is

Lines 233-235: Please verify the total number of calls given here. Summing the call totals given in the Supplementary Material results in 1253 calls, which is larger than the total given in the text before some calls were removed. Therefore, the number reported in the text should be higher, or it needs rewording for clarity.	stated 1253- there must have been an error in the calculations of the supp material somewhere. Since I have now removed call categories that had two calls per type, this number has again changed (n=1250). This has been double-checked and is consistent throughout the ms.
Comment 16 Lines 234-235: 29 categories (including 4 subtypes) matches what is shown in the call catalogue (Supplementary Material, Appendix 2) but it does not match the information in Table 2 (Supplementary Material, Appendix 1). The Table includes a 5th subtype (McM10a) that is not in the call catalogue. This needs to be changed or explained somewhere.	Thankyou for picking this up! This was an oversight. We previously did have a category 10a, but it was removed during analysis and restrictions set. It has been corrected and is removed from Table 2.
Comment 17 Lines 236-237: What is meant by “with further parameters measured” here? Aren’t these the same parameters given in Table 2 (Appendix 1)?	What was meant by “...further parameters...” was all parameters, as in the previous sentence I discuss just the summary. I have changed terminology to now state all parameters for better clarity.
Comment 18 Line 246: Call types are referred to only by number here and not with a preceding “McM” as is done later in the text. Please make this consistent.	All call types that are referred to in the text in the manuscript and appendices now have the prefix McM.
Comment 19 Would it be better to exclude the “McM” from the call types defined here when there has been a previous publication with confirmed Type C killer whale calls? It is confusing when every publication chooses their own naming scheme for call types. Perhaps this should be adding on to the findings of Schall and Van Opzeeland, 2017.	The Schall and Van Opzeeland study on Type C acoustics focused on acoustics recorded off the Eckstrom Ice Shelf- a considerable distance away from the Ross Sea. Since there is potential for geographic variation in kw vocal repertoire, it is best to keep these call types recorded in McMurdo Sound separate. Ideally if another study is done on the Type C killer whales in McMurdo Sound, this naming system of McMxx is continued.
Comment 20 Line 252: Should reference Figure 3. Comment 21 Line 263: Should reference Figure 4. Comment 22 Line 273: Should reference Figure 5.	These have all been changed accordingly. Thank you for picking this figure reference error up.
Discussion: Comment 23 Lines 355-357: Is there any information about this acoustic feature in other ecotypes in different areas to suggest it is unique to Antarctic killer whales and might be useful for identification?	I have not been able to find this acoustic feature published in other killer whale vocal literature worldwide, except for the Richlen and Schall study, which have been cited.
Comment 24 Lines 368-369: If you want to make claims like this you need to show your data. As mentioned	A table has now been added into Appendix 2 of the Supp material, giving a breakdown of every recording and number of individuals sighted per encounter. There is full transparency now on all data collected.

previously, it would be helpful to see a breakdown of the number of individuals present and behaviors documented during the encounters where recordings were made.	Claims that are not supported by corresponding graphs or statistics, such as this one, have been removed.
References: Comment 25 Formatting of references needed. The journal names are only abbreviated for some of the references.	Journal names have been doublechecked, and the abbreviations are per referencing style. I have downloaded Open Biology Royal Society referencing style from Royal Society website as per instructions for authors to RSOS. All references are formatted to this output style. Note to Editor: I feel the output style I have downloaded from Royal Society is not working. Hence I am now using Vancouver style, as per RSOS website stating, "All our journals use a system based on Vancouver style referencing".
Tables and Figures: Comment 26 Table 1: This table is unnecessary in the text and would be better suited for only the supplementary material. It seems strange to include such a lengthy table with the definitions for the parameters when none of the actual measurements are provided in the main paper. Additionally, it seems misleading to present so much information about these parameters in the methods when they were not actually used in constructing the call catalogue.	Good advice. I have now moved this table to Supplementary Material, Appendix 1, along with Table 2 – which presents the results of each measured parameter defined in Table 1. Much better!
Comment 27 Figure 6: Ideally the spectrograms in this figure could be larger as it isn't possible to read the axes labels without zooming in on the pdf. Making the axes labels clear is particularly important here since they vary between sources. Would it be possible to add larger axes labels separately?	I have edited this Figure now, so there is larger font on the axes and it is at a higher resolution as per request. The figure can be output as larger image size at editor's request- it currently stands at requested 300 dpi in TIFF format.
Comment 28 The axes for all the figures are too small to read without zooming in, as are the location labels in Figure 1. The figures are high enough resolution that you can zoom in to read these but having to zoom in on everything is not ideal. It would be better to make the axes all larger if possible.	All axis labels and spectrograms have been redone with larger font and titles for all figures. I also have done these spectrograms in grey scale if the editors would prefer this colour grade rather than the blue colour grading they stand at present. Figure 1 location labels are now bigger.
Supplementary Material: Comment 29 In Table 2, the call types are listed as numbers instead of being preceded by "McM" as they are in the text and Appendix 2. The naming scheme should be consistent throughout.	Have adjusted all accordingly- all are named with prefix McM throughout appendices and tables.

Comment 30 In Table 2, the column heading “Component” is used twice. Perhaps “Component” and then “Component Type”, where the first column has the component number and the second notes whether the component was a whistle, burst-pulse, or biphonic whistle.	Have now changed headings of both columns to 1) Component Type and 2) Component Number. Have left the columns in the same place in the table.
Comment 31 In Table 2, the n given for each call type is the total number of calls you classified for that category. This is misleading because you state in the text that you didn’t necessarily measure all of the calls for each category. Since you are presenting your parameter measurements in this table you should instead be reporting the number of calls you actually measured, especially since you don’t give exact numbers in the text for this.	I see the point the reviewer is making here, however this table needs to also display the sample size of each call type. To clear up confusion, I have modified the explanation in the Methods , where it states that a minimum of 20% of each call type were measured, and with call types with 10 or fewer examples, all calls in that category were measured.
Comment 32 The note for Table 2 should read “has measurements”.	Have changed accordingly.
Comment 33 It would make more sense to include Table 2 at the end of the call catalogue in Appendix 2. Currently, the table is repeated in both Appendices but in Appendix 2 it is broken up and does not have a legend.	There is a summary of each call type descriptive statistics included in the call catalogue. This is so both documents can be treated separately for readers- it makes it easy to compare descriptive statistics when put against a spectrogram of each call type. This was done for the reader’s benefit. Table 2 displays the full results from descriptive statistics and is quite detailed with every call category in one place – I think due to it’s large size it is best to be a separate appendix/supp material. But it is easy to merge both appendices if the editors would like this? For now I have kept them separate due to the large size and detailed information in Table 2. Supplementary Material, Appendix 1 contains Table 1 and Table 2 for consistency with other reviewer comments.
Comment 34 In Appendix 2, the description of McM1 should read “This is a 4-component and biphonic call” to be consistent with subsequent call descriptions.	Have changed accordingly.
Comment 35 In Appendix 2, the description of McM10 mentions this call being produced in a “rhythmic repeated call sequence”. Could you elaborate on what is meant by this? Is this related to there being a call type McM10a in Table 2 but not in the catalogue? What percentage of the time was this call produced in a sequence?	Call type 10a was removed from the catalogue, but mistakenly must have been left in table 2 during submission. Call type 10a was removed during analysis process. It has been fixed and is now clearly removed from the entire manuscript and supplementary material. I have also reworded the section describing this call type now, to elaborate on meaning of “repeated call sequences” and put a spectrogram on this page showing this.

If there is anything that has been missed, or anything else that needs to be addressed, please do not hesitate to contact me.

Thank you in advance.

Kind Regards,

--

Rebecca Wellard

PhD Candidate, BSc (Hons I)

Centre for Marine Science and Technology

Curtin University

GPO Box U1987, Perth, Western Australia 6845

Mobile +61 417 535 117

Email becwellard@gmail.com

Email rebecca.wellard@postgrad.curtin.edu.au

Appendix D

Dear Editors of Royal Society Open Science,

This letter is in reference to our accepted manuscript to your journal Royal Society Open Science as we present our responses to the final revisions requested by reviewers.

Manuscript ID: RSOS-191228

Title: Cold Call: The Acoustic Repertoire of Ross Sea Killer Whales (*Orcinus orca*, Type C) in McMurdo Sound, Antarctica.

Please see attached our Response to Referees.

Below are responses to each point brought up by the reviewers.

Reviewer #1 Comments and Responses

REVIEWER COMMENT	RESPONSE
Introduction Lines 106-110 – This could possibly be restructured a bit, the way it currently reads sounds like graded call are a phenomenon unique to killer whales and false killer whales, but it’s likely that it occurs in many of the odontocetes, we just haven’t done the research yet to test that. For example, there is some evidence of graded call structures in wild populations of pilot whales.	Have adjusted accordingly and included other possibly cetaceans in description. This section now reads at: Not all sounds fit into such distinct classes—whistles and burst-pulse sounds. Killer whales, as well as false killer whales (Pseudorca crassidens) and likely other cetaceans, further produce graded sound types that lie along a continuum from whistles to pulses.
Methods Lines 157-158 – note the number of encounters/recordings for each of the sampling rates. Based on your explanation, it sounds like n=1 encounter at 44 kHz sampling rate and n=8 encounters at 96 kHz, is that correct? Add a sentence here explaining how you controlled for differences in the sampling rate, or explaining why that was not necessary.	Correct. There was one field day where n=2 encounters sampled at 44kHz, this is detailed in the Supp Material (Table 1, Appendix 2). We explain how we controlled the differences in sampling rate during analysis in lines 166-168, where it states, “A fast Fourier transform was computed in Hann windows of 1024 and 512 samples with 90% overlap for the recordings sampled at 96 kHz and 44.1 kHz, respectively.” Hence, the FFT of the two sampling rates were computed differently (1024 and 512).
Discussion 367 – 167 individuals were reported in the results, should be changed here to match.	A summed total of 392 killer whales were counted over the nine encounters, although some of these individuals were re-sightings. A total of 167 distinct individuals were photographically identified from natural markings. To make this clearer for the reader, it now reads at, “concurrent visual and acoustic recordings that confirmed the ecotype over nine encounters with a summed total of 392 individuals.”

Reviewer #2 Comments and Responses

REVIEWER COMMENT	RESPONSE
Line 52: First figure mention – numbering should start with “1” here, suggest changing the order.	Have removed this reference to Figure 4, it was out of place and unnecessary. Thanks for picking this up.
Line 91-92: The reference “42” talks about pulsed calls, not amplitude-modulated whistles. This needs to be revised. Not all amplitude modulated whistles have sidebands but only those that actually contain pulses. You can have an amplitude modulated continuous wave without sidebands but harmonics.	As Murray showed, calls of false killer whales lie along a continuum from whistles to pulses. This continuum is achieved by progressively increasing amplitude-modulation, with the extreme case, where the amplitude is modulated so much that series of pulses appear. So pulsed tones, or series of tone pips, can be seen as resulting from extreme amplitude-modulation. Note Murray went one step further and increased the inter-pulse duration after single pulses had been formed. Also note that amplitude-modulation (unless you also change the amplitude of the modulating frequency) does not affect the peak amplitude of the underlying (carrier) sinusoid, but only imposes an envelope that creates lower sinusoid amplitudes in between the peaks of the waveform. Amplitude-modulation causes sidebands. That’s the fundamental concept underlying communication theory. If the amplitude is modulated by a subharmonic of the carrier frequency, then the call spectrogram will appear to have harmonics. The reviewer is correct that ref 42 does not discuss amplitude-modulation but rather different degrees of pulsing a tone (i.e., removing more and more cycles as pulse-repetition decreases). We’ve added a general reference for amplitude-modulation creating sidebands in communication theory (see Faruque 2017).
Line 93-94: I don’t think you can throw the ultrasonic whistles/high-frequency modulated signals into the same description as relatively low-frequency “standard” whistles. That slipped my notice last time around. These are two distinctly different categories and very separate from each other in frequencies as well as likely behavioral use.	This section is about the broad characteristics of the broadest classes of sounds that orca make: burst-pulse sounds, whistles, and clicks. Some authors have found low-frequency whistles and high-frequency whistles. There are papers on both, or one or the other. There does not seem to be a clear cut-off frequency, below which whistles are considered low-frequency and above which whistles are consistently considered high-frequency. In fact, in our data, whistles lay along yet another continuum, this time in frequency; meaning whistles ranged over a broad range of frequencies with no frequency band demarcating LF whistles and HF whistles. Therefore, in this paragraph, we generally describe the range of frequencies occurring in whistles. We discuss the fact

	that we did not find LF versus HF classes further down in our manuscript. We have not edited lines 93-94.
Line 108-109: It seems that the term amplitude-modulation is used interchangeably with pulse-repetition rate. That doesn't seem appropriate here or in lines 91-92.	Conceptually, thinking about graded sound types, coming from the whistle end of the continuum, you can approximate pulses by strong amplitude-modulation; and thereafter, pulse series can be created by increasing the inter-pulse interval. So in this concept, we see amplitude-modulation to pulsing along a continuum. To get to a series of distinct pulses, increases in inter-pulse interval are necessary (as shown by Murray) and we've clarified this in line 108-109. It now reads at, "This continuum is achieved by progressively increasing the amplitude-modulation of whistles until successive pulses are formed, and increasing the inter-pulse interval thereafter.
Line 167-168: What was the time window? And what was the size of the window on your screen? Ultimately, if you manually pick your measurements then your time/frequency precision is only partially limited by your FFT and overlap but also by the number of pixels available on your screen to display that resolution.	We've added the time window to line 168. The spectrograms were displayed at a resolution much greater than the FFT frequency resolution, making the FFT the limiting factor. This now reads at, "A fast Fourier transform was computed in Hann windows of 1024 and 512 samples with 90% overlap for the recordings sampled at 96 kHz and 44.1 kHz, respectively, resulting in a frequency resolution of about 90 Hz and time window of about 11 ms."
Line 365: "comprehensive" – this is a strong statement; it's a large repertoire but may not be comprehensive (as indicated in discussion).	The term comprehensive is not referring to the size of the repertoire, but rather the study and the thorough analysis of calls and methodology. Reference to the size of the repertoire is in lines 530-532.
Figure 1: Map labels (axes and other text) too small.	We have edited labels on map and axes and increased font size.
Figures 2-8: the actual frequency and time values are still way too small, especially in figure 8.	We have edited spectrogram figures 5-7 and increased font size.
Figure 8: Comparative graphs should be scaled in time and frequency such that they are actually comparable and legible.	We have adjusted the axes on every spectrogram in this comparative figure, to make it more legible. Unfortunately, we don't have access to the raw data of these wav files (we did try to obtain raw data files). We only have the images of the spectrograms created by others, of which we adjusted the axes, but could not change the entire spectrogram.
Appendix 2: Table 1 – would it be possible, maybe instead of column bit depth, which doesn't change and isn't too important anyways, to include recording duration?	We have added a column of recording durations for every encounter into Table 1, Appendix 2. We were able to keep the column of bit depth as well.

If there is anything that has been missed, or anything else that needs to be addressed, please do not hesitate to contact me.

Thank you in advance.

Kind Regards,

--

Rebecca Wellard

PhD Candidate, BSc (Hons I)

Centre for Marine Science and Technology

Curtin University

GPO Box U1987, Perth, Western Australia 6845

Mobile +61 417 535 117

Email becwellard@gmail.com

Email rebecca.wellard@postgrad.curtin.edu.au